# Estimating the variability of $NO_x$ emissions from Wuhan with TROPOMI $NO_2$ data during 2018 to 2023

Qianqian Zhang[1], K. Folkert Boersma[2,3], Chiel van der Laan[4], Alba Mols[2], Bin Zhao[5,6], Shengyue Li[5,6], Yuepeng Pan[7,8]

[1]National Satellite Meteorological Center, Key Laboratory of Radiometric Calibration and Validation for Environmental Satellites, Innovation Center for Fengyun Meteorological Satellite (FYSIC), China Meteorology Administration, Beijing 100081, China
[2]Wageningen University, Environmental Science Group, Wageningen, the Netherlands
[3]Royal Netherlands Meteorological Institute, De Bilt, the Netherlands
[4]Eindhoven University of Technology, Eindhoven, the Netherlands
[5]State Key Joint Laboratory of Environmental Simulation and Pollution Control, School of Environment, Tsinghua University, Beijing 100084, China
[6]State Environmental Protection Key Laboratory of Sources and Control of Air Pollution Complex, Beijing 100084, China.
[7]Key Laboratory of Atmospheric Environment and Extreme Meteorology, Chinese Academy of Sciences, Beijing 100029, China
[8]College of Earth and Planetary Sciences, University of Chinese Academy of Sciences, Beijing 100049, China

*Correspondence to*: Qianqian Zhang, zhangqq@cma.gov.cn

**Abstract.** Accurate $NO_x$ emission estimates are required to better understand air pollution, investigate the effectiveness of emission restrictions, and develop effective emission control strategies. This study investigates and demonstrates the ability and uncertainty of the superposition column model in combination with the TROPOspheric Monitoring Instrument (TROPOMI) tropospheric $NO_2$ column data to estimate city-scale $NO_x$ emissions and chemical lifetimes and their variabilities. Using the recently improved TROPOMI tropospheric $NO_2$ column product (v2.4–2.6), we derive daily $NO_x$ emissions and chemical lifetimes over the city of Wuhan for 372 full-$NO_2$-coverage days between May 2018 and December 2023, and validate the results with bottom-up emission inventories. We find insignificant weekly cycle of $NO_x$ emissions for Wuhan. We estimate a summer-to-winter emission ratio of 0.77, which is overestimated to some extent, though it is even higher provided by the bottom-up inventories. We calculate a steady decline of $NO_x$ emissions from 2019 to 2023 (except for the valley in 2020 caused by the COVID-19 lockdown), indicating the success of the emission control strategy. The superposition model method results in ~15% lower estimation of $NO_x$ emissions when the wind direction is from distinct upwind $NO_2$ hotspots compared to other wind directions, indicating the need to improve the approach for cities that are not relatively isolated pollution hotspots. The method tends to underestimate $NO_x$ emissions and lifetimes when the wind speed is $> 5-7$ m s$^{-1}$, the estimation for Wuhan is ~4% for the emissions and ~8% for the chemical lifetime. The results of this work nevertheless confirm the strength of the superposition column model in estimating urban $NO_x$ emissions with reasonable accuracy.

# 1 Introduction

Nitrogen oxides ($NO_x \equiv NO_2 + NO$) are key atmospheric components affecting the formation of particulate matter and ozone (Bassett and Seinfeld, 1983; Jacob, 1999; Penner et al., 1991). They are emitted into the atmosphere mainly from the combustion of fossil fuels, which takes place primarily in urban areas, to heat and provide electricity to homes and businesses and to run cars and factories. Cities are responsible for more than 70% of global $NO_x$ emissions, and this proportion increases with the process of global urbanization and industrialization (Park et al., 2021; Stavrakou et al., 2020; Baklanov et al., 2016). Thus, accurate $NO_x$ emission inventories for cities are required for monitoring the effectiveness of reducing air pollution and for global and regional chemical models to reproduce the complicated urban air pollution (Beirle et al., 2011). Bottom-up city $NO_x$ emission inventories are quite uncertain in the emission factors (Lu et al., 2015) and during the down-scaling from national or regional emissions to city level (Butler et al., 2008; Lamsal et al., 2011).

NO₂ has long been detected by remote sensing instruments with high quality because of its strong spectral features within the ultraviolet (UV)/visible spectrum. Various satellite instruments have been providing tropospheric NO₂ column measurements for near-surface $NO_x$ emissions estimation for tens of years (Burrows et al., 1999; Bovensmann et al., 1999; Levelt et al., 2006; Veefkind et al., 2012). Limited by the coarse spatial resolution of the early instruments, researchers computed the global or regional long-term mean $NO_x$ emissions with satellite observations and chemical transport models (CTMs) (Martin et al., 2003; Lamsal et al., 2011; Kharol et al., 2015). With the improving capabilities of later satellite sensors, more researchers started to estimate $NO_x$ emissions on higher spatial and temporal resolutions but still depended on CTMs (e.g., Ding et al., 2017; Visser et al., 2019; Xing et al., 2022). However, there are barriers to access and employment of CTMs, and there is a substantial computational burden when our target is an individual city. Therefore, CTM-independent methods have been developed and applied to estimate $NO_x$ emissions since the early 2010s (e.g., Beirle et al., 2011; De Foy et al., 2014; Kong et al., 2019; Beirle et al., 2019; Rey-Pommier et al., 2022).

Beirle et al. (2011) reduced the 2-dimensional NO₂ map surrounding a large point source (such as a megacity or a power plant, factory) to the 1D NO₂ line density by integrating the NO₂ column density across the wind direction. They developed an Exponentially Modified Gaussian method (EMG) to estimate $NO_x$ emissions and lifetime from the increase of NO₂ over the source and its decay downwind of the city. Over the years, the model has been refined (Valin et al., 2013), validated (De Foy et al., 2014; 2015), applied (e.g., Lu et al., 2015; Lange et al., 2022; Goldberg et al., 2019) and extended (Liu et al., 2016; 2022). The EMG method has been first applied to OMI NO₂ data to calculate $NO_x$ emissions from cities, power plants, and factories across the globe and was shown to be accurate when wind speed is larger than 2 or 3 m s$^{-1}$. This method is frequently used to calculate mean $NO_x$ emissions over longer time periods (like some years) with OMI data (e.g., Lu et al., 2015; Liu et al., 2016).

At present, the much improved spatial resolution and retrieval quality of the TROPOMI sensor allows the quantification of episodic (like biomass burning) or even daily $NO_x$ emissions with the EMG method (Goldberg et al., 2019; Jin et al., 2021). Based on a single TROPOMI overpass, Lorente et al. (2019) developed a superposition column model to fit the NO₂ line

density for daily NO$_x$ emissions. They found highest emissions on cold weekdays and lowest emissions on warm weekend days, indicating the significant contribution from home heating during winter in Paris. Zhang et al. (2023) used this model to estimate the daily variation of NO$_x$ emissions over Wuhan from September 2019 to August 2020, and further inferred CO$_2$ emissions based on the simultaneous and co-located NO$_2$ and CO$_2$ satellite observations. The superposition column model estimates NO$_x$ chemical lifetimes and emissions on daily basis, avoiding the bias caused by using the averaged NO$_2$ columns

in the nonlinear system (Valin et al., 2013). However, Lorente et al. (2019) and Zhang et al. (2023) used daily hydroxyl radical (OH) concentration from CTMs as an important parameter, which induced computational burden when the study period is as long as several years. In addition, the CTM output OH concentration is highly uncertain (Zhang et al., 2021) and leads to uncertainty in the estimated city NO$_x$ emissions and chemical lifetime.

In this study, we continue to focus on the city of Wuhan, extend our study period from May 2018 to December 2023, and

estimate city NO$_x$ emissions and lifetimes on a daily basis with the superposition column model. We discard CTM output OH concentration in the method to reduce the results' uncertainty and improve computing efficiency. Our purpose is, first, to demonstrate the ability of the superposition column model to provide information on city NO$_x$ emissions and lifetimes on interannual, seasonal, and weekly variations influenced by changes in human activity; second, to investigate the model performance influenced by the meteorology (wind speed and directions). The rest of the paper is organized as follows:

Section 2 introduces the data and methods we employ in this study. In Sect. 3, we compare our results with those of other studies, analyze the temporal variability of NO$_x$ emissions and lifetimes over Wuhan, investigate the dependence of our estimations on the wind field. We discuss the uncertainties of this work in Sect. 4. The concluding remarks are presented in Sect. 5.

## 2 Data and Material

### 2.1 TROPOMI NO$_2$ tropospheric columns

On 13 October 2017, the Copernicus Sentinel-5 Precursor (S-5P) satellite was successfully launched into a sun-synchronous orbit with the local overpass time at around 13:30. The TROPOspheric Monitoring Instrument (TROPOMI) is the only instrument on board S-5P, dedicated to air quality and climate monitoring. TROPOMI NO$_2$ columns are retrieved in the spectral range from 405 to 465nm of the UV-visible spectral band with a nadir spatial resolution of 7.2 $\times$ 3.6 km$^2$ (reduced

to 5.6 $\times$ 3.6 km$^2$ as of 6 August 2019) (Van Geffen et al., 2020; 2022). The small pixels and large swath width (approximately 2600 km) of TROPOMI allow the detection of localized point sources and downwind NO$_2$ plumes from cities on a daily basis (Beirle et al., 2019; Lorente et al., 2019).

This work uses the operational TROPOMI NO$_2$ version 2.4.0–2.6.0 algorithm from May 2018 to December 2023. Compared to the previous versions v1.x, the version 2.3.1 includes a different treatment of the surface albedo to avoid

negative and > 1 cloud fractions, and updates the FRESCO-wide cloud retrieval that leads to a lowering cloud pressure.

These result in 10%－40% increase of tropospheric $NO_2$ columns, depending on the level of pollution and season (Van Geffen et al., 2022; Lange et al., 2023). There is a major change from version 2.3.1 to 2.4.0. In v2.4.0, a new TROPOMI surface albedo climatology (Directional Lambertian Equivalent Reflectivity, DLER) was implemented in the cloud fraction and cloud pressure retrievals and air-mass factor calculation (Eskes et al., 2023). The use of the DLER results in a substantial increase of $NO_2$ columns in vegetated regions, and the higher resolution ($0.125° \times 0.125°$) of the DLER better resolves the variability in the surface albedo (Keppens and Lambert, 2023). The version 2.4.0 made a complete mission reprocessing from 1 May 2018 to 22 July 2022 and then switched to the offline mode. The version 2.5.0 implemented a minor bug fix concerning the qa_value field over snow or ice-covered regions. The version 2.6.0 started on 16 November 2023, and is exactly the same as version 2.5.0.

The version 2.4.0–2.6.0 Level 2 tropospheric $NO_2$ products are reported to be biased between +33% (over cleaner areas) and –50% (over highly polluted areas) compared to the ground-based MAX-DOAS data from 29 stations, with the overall negative median bias being 28%. Amongst all the 29 ground stations, one is in north China, Xianghe. At Xianghe, the TROPOMI tropospheric $NO_2$ columns correlate ($R^2 = 0.88$) well with the MAX-DOAS data with a median low bias of ~20%, and the weekly averaged relative difference is within ±30% for most of the days (Keppens and Lambert, 2023). When we use the TROPOMI data in Wuhan, some low bias is expected, and we adjust a scale factor of 1.2 to (partly) correct the bias and screen each ground pixel for the quality assurance flag (qa_value) greater than 0.75.

## 2.2 Wind data

Apart from the tropospheric $NO_2$ columns, wind field (wind direction and wind speed) is needed as forward model parameters to determine city $NO_x$ emissions and chemical lifetimes. We use the reanalysis wind data on pressure levels provided by the fifth-generation (ERA5) European Center for Medium-Range Weather Forecasts (ECMWF) (Hersbach et al., 2020). This dataset provides hourly wind data on 37 vertical levels with the horizontal resolution of $0.25° \times 0.25°$. Considering the vertical consistency of wind speeds and directions, we use the 3 levels mean meridional and zonal wind below 950hPa. The two hourly values immediately before and after the TROPOMI overpass timestamp are linearly interpolated.

## 2.3 Emission inventory

An initial guess of $NO_x$ emission patterns and amounts are needed in this study, and we use the Air Benefit and Attainment and monthly Cost Assessment System Emission Inventory (ABACAS-EI) (Zhao et al., 2013; Zhao et al., 2018; Zheng et al., 2019) for the year 2019 to provide this information. The ABACAS-EI is available at 1km $\times$ 1km gridded resolution over China, and it is developed based on activity rates and energy consumption levels with an estimated uncertainty of ±35% (Zhao et al., 2013; Li et al., 2024b). Two other bottom-up $NO_x$ emission inventories are also used for comparison. The first one is the Emission Database for Global Atmospheric Research (EDGAR) v8.1, which provides monthly sectoral $0.1° \times 0.1°$

NO$_x$ emissions from 2000 to 2022, and the 2018－2022 data are employed. The second is the 2018－2020 monthly NO$_x$ emission from the Multi-resolution Emission Inventory model for Climate and air pollution research (MEIC) v1.4, with the spatial resolution of 0.25 °×0.25 °(Zheng et al., 2021a; 2021b).

## 2.4 The superposition column model

Lorente et al. (2019) introduced this superposition column model to estimate city NO$_x$ emissions and chemical lifetimes based on a single TROPOMI overpass and determined daily NO$_x$ emissions over Paris. Zhang et al. (2023) modified and used this model for the Chinese city of Wuhan in a more polluted background.

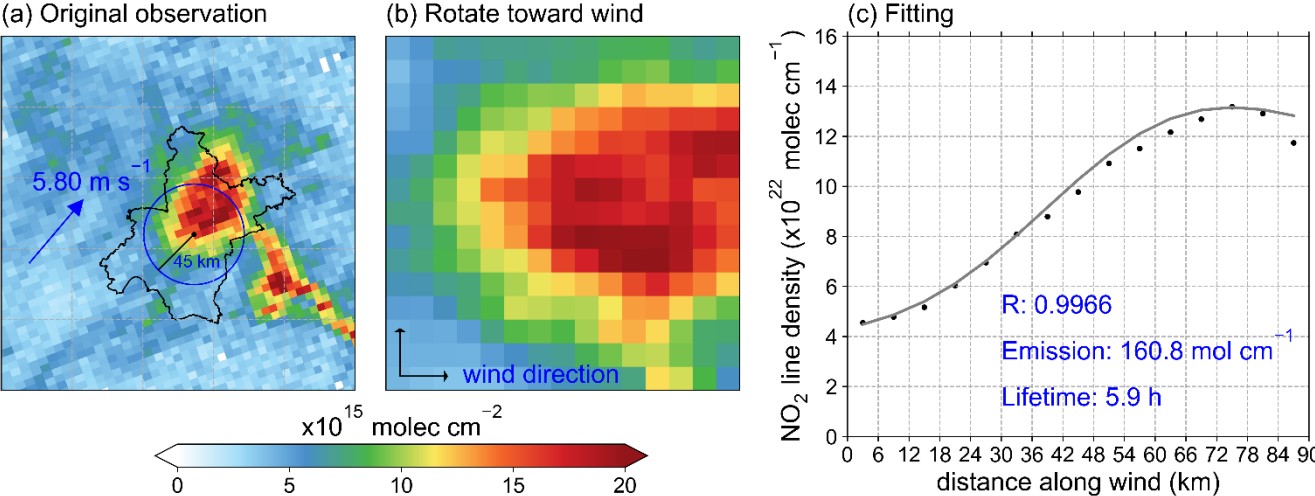

**Figure 1: The analysis steps of the superposition column model. (a) The original TROPOMI NO₂ columns are on 4 March 2023. The administrative boundary of Wuhan is plotted, and the blue circle inside denotes the study domain centered on Wuhan city center (114.4 ℉E, 30.6 ℉N) with a 45 km radius. (b) NO₂ columns resampled at a grid map aligned with the wind direction within the study domain. (c) The grid cells perpendicular to the wind direction are integrated to obtain the NO₂ line density and fit it with the superposition column model.**

Using the day 4 March 2023 as an example, we demonstrate the analysis steps of the superposition column model (Fig. 1). We focus on the build-up of NO₂ columns over a 45 km radius around the city center 114.4 ℉E, 30.6 ℉N, which covers the urban area of Wuhan (the blue circle in Fig. 1a). Compared to Zhang et al. (2023), our study domain is limited to the urban area (within the Fourth Ring Road) of Wuhan. For one reason, most (~ 60%) of the NO$_x$ emissions are concentrated in the urban area (Zhang et al., 2023); for another, we use regional mean wind fields and NO$_x$ chemical loss rate, the larger study domain would induce large uncertainty in the result. We construct a 15 × 15 grid map centered at the city center with each grid size of 0.05 °× 0.05 °(6km × 6km) toward the mean wind direction. One demission of the grid map is along with and the other perpendicular to the wind direction. The mean wind direction is determined by the mean meridional and zonal winds

over the study domain. The original TROPOMI observation (Fig. 1a) is sampled into the rotated grid map (Fig. 1b). The TROPOMI NO$_2$ columns in the 15 grid cells perpendicular to the wind direction are integrated to form the so-called NO$_2$ 'line densities' (Beirle et al., 2011), resulting in 15 grid cells along the wind direction (Fig. 1c).

Then, the NO$_2$ line density is fitted with the superposition column model (Lorente et al., 2019; Zhang et al., 2023) which is based on a simple column model (Jacob, 1999). We solve each grid cell along the wind direction with the simple column model, NO$_2$ builds up within the current cell and decays exponentially downwind of this cell.

$$N_i(x) = \frac{E_i}{k}\left(1 - e^{-kL/u}\right) \cdot e^{-k(x-x_i)/u} \cdot \frac{1}{[NO_x]/[NO_2]} \quad for\ x > x_i \quad , \tag{1}$$

$$N_i(x) = 0 \qquad\qquad\qquad\qquad for\ x \leq x_i\ , \tag{2}$$

$$N(x) = \sum_{i=1}^{n} N_i(x) + b + \alpha x \tag{3}$$

NO$_x$ emissions from cell $i$ ($E_i$, mol cm$^{-1}$ s$^{-1}$) along the wind direction contribute ($N_i(x)$, mol cm$^{-1}$) to the overall line density through the build-up of NO$_2$ within the current cell and exponential decay in the downwind cells (Eq. (1)). We assume a first-order loss of NO$_2$ in the atmosphere. In Eq. (1), $k$ (s$^{-1}$) represents the loss rate of NO$_2$ at the TROPOMI overpass time, and the relationship between $k$ and NO$_2$ chemical lifetime $\tau_{[NO_2]}$ (h) is $k = \frac{1}{\tau_{[NO_2]}*3600}$. Here, we make an initial guess to the NO$_2$ chemical lifetime of 4 h for cold months (October to March) and 2 h for warm months (April to September) to derive the parameter $k$. We set a large range for $k$ to allow it to change between 1/4 to 4 times of the initial value. $L$ denotes the length of each grid cell, i.e. $6 \times 10^5$ cm, and $u$ is the wind speed (cm s$^{-1}$). We follow Zhang et al. (2023) to take a fixed value 1.26 for $[NO_x]/[NO_2]$. $E_i$ makes no contribution to its upwind cells (Eq. (2)).

The contributions from all the 15 cells are stacked up to construct the superposition column model, and combined with the contribution from the background NO$_2$ line densities ($b + \alpha x$) to obtain the overall $N(x)$ (Eq. (3)). The initial guess of the background value $b$ is set as the NO$_2$ line density at the upwind end point $N(0)$.

The terms $E_i$, $k$, $\alpha$ and $b$ are fitted through a least squares minimization to the TROPOMI observed NO$_2$ line densities ($N_{TROPOMI}(x)$) and the a priori ABACAS NO$_x$ emissions ($E_{ABACAS,i}$) to determine $N(x)$. The cost function is defined as follows:

$$func = \left(\frac{N(x)-N_{TROPOMI}(x)}{N_{TROPOMI}(x)}\right)^2 + fac * \left(\frac{E_i - E_{ABACAS,i}}{E_{ABACAS,i}}\right)^2 \tag{4}$$

The emission term is used in the cost function to reduce the dependence of fitted NO$_x$ chemical lifetimes and emissions on the $\tau_{[NO_2]}$, for too large a varying range is applied for it. We exert a scale factor ($fac$) changing between 0.1 and 0.2 to the emission term to make sure the cost function is dominated by the NO$_2$ line density term.

Finally, the total NO$_x$ emissions $E$ (in the unit of mol s$^{-1}$) from the study domain can be calculated with

$$E = \sum_{i=1}^{15} E_i \times L \tag{5}$$

and the estimated NO$_x$ ($\tau_{[NO_x]}$) chemical lifetime is obtained through Eq.(6) (Seinfeld and Pandis, 2016):

$$\tau_{[NO_x]} = \frac{1}{k \cdot 3600} \cdot \frac{[NO_x]}{[NO_2]} \qquad (6)$$

## 3 Results

### 3.1 Mapping Wuhan's $NO_x$ emissions and chemical lifetimes

From May 2018 until December 2023, we collect 581 overpasses with full-$NO_2$-coverage over Wuhan. We remove the overpasses with inhomogeneous wind fields, which happen most frequently in winter. The inhomogeneous wind fields include the situations when the wind direction changes more than $45°$ within 2 hours before the satellite overpasses, and the wind at zonal or meridional direction reverses at different pressure levels when the satellite overpasses. Multiple overpasses within one day are analysed separately to calculate $NO_x$ emissions and then averaged for the daily mean emission level.

Finally, we obtain a total of 372 days with valid $NO_x$ emissions and chemical lifetimes estimations. The number of valid days for each year and each season are summarized in Table 1. For the five years (2019－2023) with full-year measurement, the percentage of days with valid estimations is 15.3%－22.5%. Seasonally, we obtain the most valid days in autumn (defined as September to November), followed by summer (June to August). There are least valid days in winter (December to February) due to the cloudy and polluted conditions.

**Table 1: Number of days with valid $NO_x$ emissions and lifetime estimations for each year and each season.**

| By year | 2018 | 2019 | 2020 | 2021 | 2022 | 2023 |
|---|---|---|---|---|---|---|
|  | 28 | 69 | 56 | 65 | 82 | 72 |
| By season[a] | Spring | | Summer | | Autumn | | Winter |
|  | 82 | | 70 | | 131 | | 62 |

a. The COVID-19 lockdown-influenced days (23 January to the end of April in 2020) are not considered when we count the valid days by season.

### 3.1.1 $NO_x$ emissions

Zhang et al. (2023) used the superposition column model to calculate $NO_x$ emissions over Wuhan for the period from September 2019 to August 2020. They calculated 11.5 kg s$^{-1}$ (equivalent to about 250 mol s$^{-1}$) $NO_x$ emissions over Wuhan (including the urban area and the outskirts of Wuhan) from September to November 2019. We estimate $148.8 \pm 48.8$ mol s$^{-1}$ $NO_x$ emissions over the urban region of Wuhan for the same period, indicating ~60% of $NO_x$ emissions are concentrated over the central area. Lange et al. (2022) applied the EMG method to calculate $NO_x$ emissions over Wuhan from March 2018 to January 2020 of $115.1 \pm 10.1$ mol s$^{-1}$, and they used much earlier TROPOMI data versions 1.1.0－1.3.0 with larger low bias and they did not apply an ad hoc correction factor of 1.2 as we do in this study.

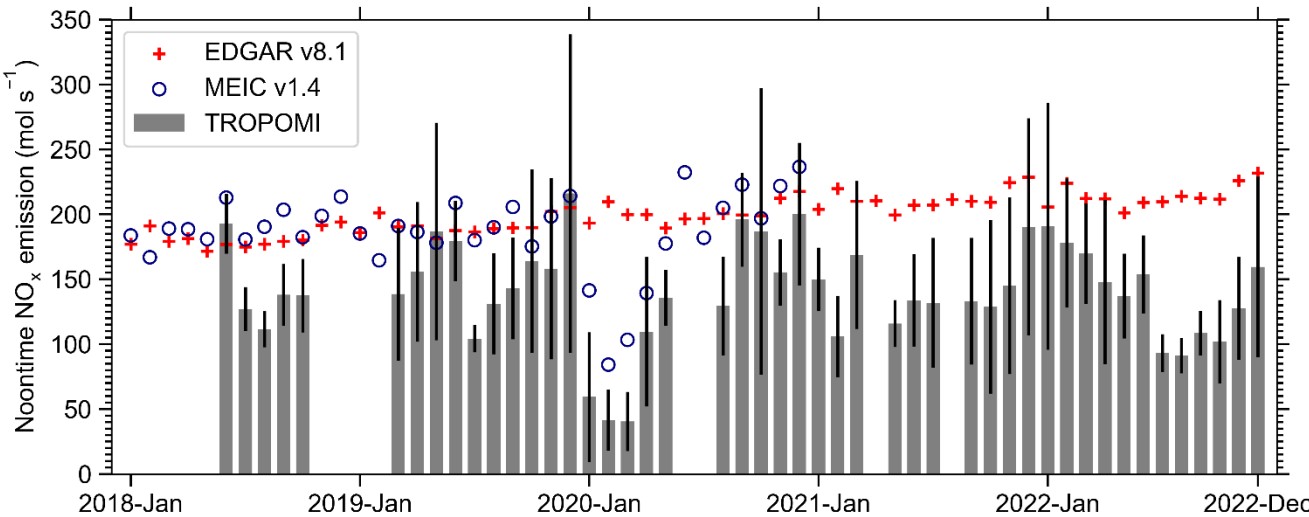

**Figure 2: Monthly NO$_x$ emissions estimated with the TROPOMI data (grey columns), reported by EDGAR v8.1 (red pluses) and MEIC v1.4 (blue circles) from 2018 to 2022 (from 2018 to 2020 for MEIC v1.4). The emissions of EDGAR v8.1 and MEIC v1.4 are scaled to the noontime emission intensities to keep in consistent with TROPOMI. The TROPOMI monthly mean is calculated only when three or more days are available. Thus, the comparison is unavailable for several months. The error bars on the TROPOMI estimations represent the standard deviation of the daily NO$_x$ emissions in each month.**

We now compare the NO$_x$ emissions over Wuhan estimated in this study with those from EDGAR v8.1 and MEIC v1.4 in Fig. 2. The monthly total NO$_x$ emissions are provided in the EDGAR v8.1 and MEIC v1.4, and we convert them into monthly mean noontime emission intensities with the time factor in ABACAS-EI. Since the monthly mean TROPOMI NO$_x$ emissions are calculated only when three or more days' NO$_x$ emissions are available, the comparison is missing for the months November 2018 to February 2019, June and July 2020, April and August 2021. Overall, the TROPOMI estimation is close to the bottom-up emission inventories during cold months, while much lower during warm months. For the three years (2018 to 2020) when MEIC v1.4 data is available, the difference between TROPOMI and MEIC v1.4 is within 35%, and both of them capture the NO$_x$ emission reduction in early 2020 due to COVID-19 lockdown. TROPOMI and EDGAR v8.1 are close to each other (within 30% difference) in 2018 and 2019, but the discrepancy is larger since 2020. EDGAR v8.1 is > 50% higher than TROPOMI from 2020 to 2022.

### 3.1.2 NO$_x$ chemical lifetimes

Different from Lorente et al. (2019) and Zhang et al. (2023), we do not use the CTM simulated daily OH concentration to constrain city NO$_x$ chemical lifetime, instead we fit it around an initial value. The initial guess of the chemical lifetime for cold and warm months are determined according to the fitting results of Zhang et al. (2023).

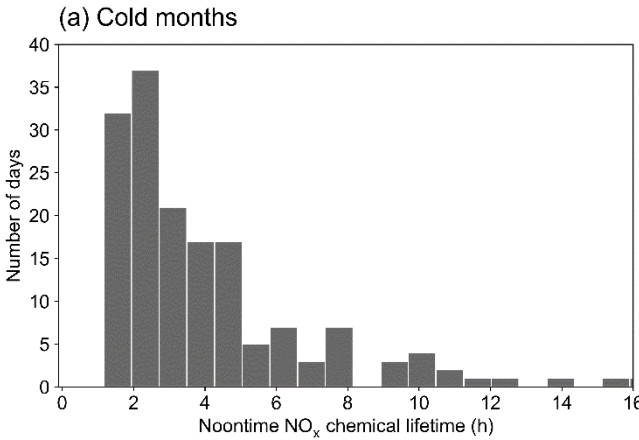
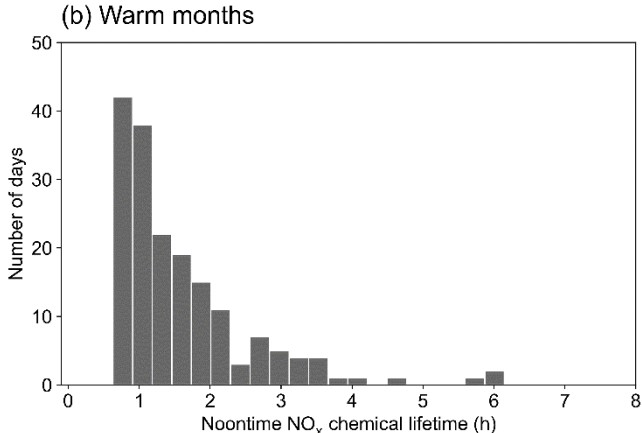

**Figure 3: Distribution of estimated NO$_x$ chemical lifetimes over Wuhan in (a) cold months and (b) warm months.**

The final generated NO$_x$ chemical lifetime over Wuhan is displayed in Fig. 3. Overall we calculate a mean NO$_x$ chemical

lifetime of 2.82 h, close to the 2.46 h estimated by Zhang et al. (2023), and is around 5% lower than Lange et al. (2022) reported 2.94±0.3 h for the NO$_x$ effective lifetime. The fitting result for cold months NO$_x$ chemical lifetime is 4.25 h, and for most of the days, the estimated NO$_x$ chemical lifetime is between 1.5 h and 6 h. For the warm months, most of the estimates are within the 0.8 – 2.5 h range, and the mean value is 1.62 h.

### 3.2 Temporal variability of NO$_x$ emissions over Wuhan

Considering the atmospheric photochemical activity of NO$_x$, previous studies using the build-up of NO$_2$ pollution along the wind direction are primarily based on satellite NO$_2$ data in the warm months (Lu et al., 2015; Liu et al., 2016; Goldberg et al., 2021b). Lange et al. (2022) have proved that it is also possible to fit the NO$_2$ line densities for NO$_x$ emissions and lifetimes in winter when NO$_x$ lifetimes are much longer. Zhang et al. (2023) estimated a year-round daily NO$_x$ emissions and lifetimes over Wuhan from September 2019 through August 2020. In this study, we extend the study period to 6 years. We have more

than 40 valid days for each day of the week, and more than 60 days for each season and each year (2018 excepted), making it possible to investigate the time variabilities of NO$_x$ emissions over Wuhan on the weekly, seasonal, and interannual scales.

### 3.2.1 Weekly cycle

To identify the weekly cycle of NO$_x$ emissions over Wuhan, we exclude the 27 valid days during the COVID-lockdown period, resulting in 345 valid daily NO$_x$ emission estimates. The weekend effect (defined as the reduction in NO$_2$ columns or

NO$_x$ emissions on weekends compared to weekdays) is widely recognized and reported in cities around the world but not in Chinese cities (Beirle et al., 2003; Stavrakou et al., 2020; Zhang et al., 2023; Goldberg et al., 2021a). Beirle et al. (2003) explained the absence of a weekend effect in Chinese cities by the dominant role of power plants and industry in NO$_x$ emission sources. Stavrakou et al. (2020) found a slight reduction of NO$_2$ columns on weekends compared to the weekday

average in Chinese cities from 2005 to 2017. Because of China's clean air action on power plants and the growing vehicle population, transportation has replaced power plants as the dominant contributor to $NO_x$ emissions. However, we still find no significant reductions on weekends for Wuhan, as shown in Fig. 4. Wei et al. (2022) also reported weak weekday/weekend differences in surface $NO_2$ concentration over Chinese cities. The 'Annual Report on Wuhan Transportation Development (2023)' (https://jtj.wuhan.gov.cn/znjt/zxdt/202409/t20240904_2450210.shtml, last access: 25 November 2024, in Chinese) revealed that the traffic flow passed through the Outer Ring Road and the Fourth Ring Road of Wuhan was highest on Friday and lowest on Tuesday and Sunday, but the difference is only less than 2%. Our result also sees an insignificant maximum on Friday and minimum on Tuesday. Cultural and living differences with other cities might explain the absence of a strong weekend effect in Wuhan. Shops, restaurants, and traffic are much busier on weekends, especially at noon when the satellite passes. Lange et al. (2022) reported a 0.79 weekday-to-weekend ratio for Wuhan, the possible reason is that our study is limited to the urban area, while a larger area is needed by Lange et al. (2022) to perform the EMG method. The different behaviors and sources in the urban and suburban areas might lead to the different weekend-to-weekday emission ratio.

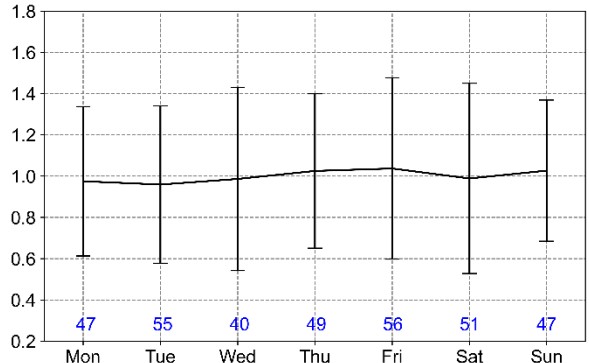

**Figure 4: Weekly cycle of mean (2018－2023) $NO_x$ emissions. The emissions are normalized with respect to the mean emissions of all the days. The number of valid measurement days for each day of the week is listed in the plot. The error bars represent the standard deviation of the daily $NO_x$ emissions.**

### 3.2.2 Seasonal pattern

Seasonal $NO_x$ emissions over Wuhan are plotted in Fig. 5a. Our results reveal that Wuhan $NO_x$ emissions vary from season to season with the highest emission in winter and lowest in summer. Winter emissions are higher by 9.6%－29.4% than the other three seasons. This is different from the bottom-up emission inventories which show little seasonal difference, as shown in Fig. 2. According to the bottom-up emission inventories, a small seasonal variation of $NO_x$ emissions should be expected, for transportation and industry are the two dominant contributors to $NO_x$ emissions over Wuhan, making up nearly 90% of the total $NO_x$ emissions, and these two emission sectors exhibit no significant seasonal variations (Zheng et al., 2018). Also, Wuhan is located in central China where winter is mild, and there is no domestic heating in winter. The summer-to-winter $NO_x$ emission ratio usually serves to indicate the relative importance of winter heating and summer power

consumption due to air conditioning. Here we calculate a summer-to-winter ratio of 0.77, and Lange et al. (2022) reported an
even lower 0.3, while it is larger than 0.85 in the bottom-up emission inventory.

Two possible factors may contribute to the large difference in the summer-to-winter emission ratio between this study and Lange et al. (2022). First is the different treatment to the $NO_x$-to-$NO_2$ ratio. We use a fixed $NO_x$-to-$NO_2$ ratio of 1.26, while Lange et al. (2022) calculated the ratio from day to day, and it was lower in summer than in winter, leading to a lower $NO_x$ emission estimation in summer. Second is that we use the bottom-up emission inventory to constrain our estimation, the flat
seasonality of the bottom-up emissions leads to a higher summer-to-winter ratio of this study.

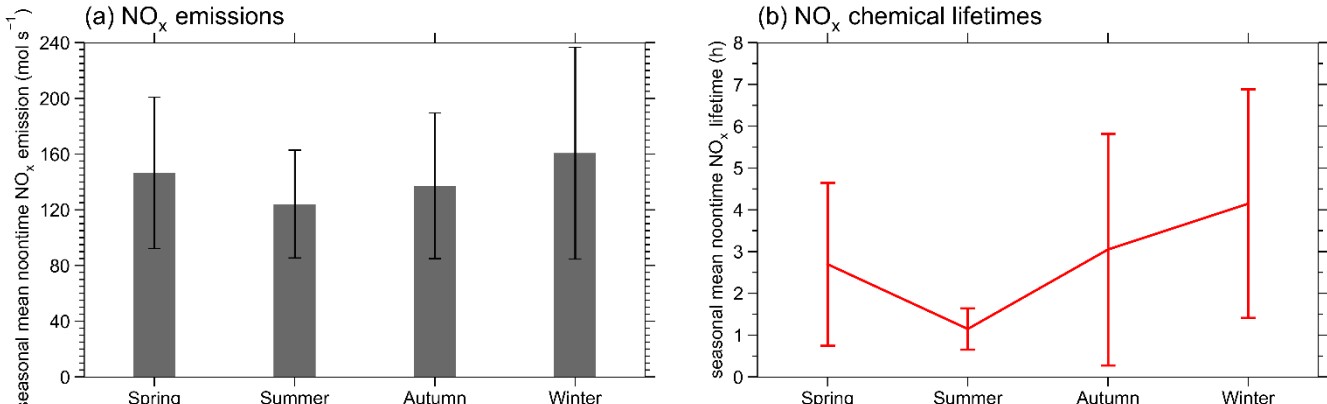

**Figure 5: TROPOMI estimated seasonal mean noontime (a) $NO_x$ emissions and (b) chemical lifetimes over Wuhan. The error bars represent the standard deviation.**

Dominated by the $NO_x$ photochemical activity rate under the influence of temperature and OH concentrations, the $NO_x$
chemical lifetimes are longest in winter and shortest in summer, as shown in Fig. 5b. Spring and autumn $NO_x$ chemical lifetimes are close to each other between winter and summer. For winter, we estimate an average of 4.1 h, close to the 4.8 h reported by Zhang et al. (2023), and longer than the ~3 h effective lifetime estimated by Lange et al. (2022), which makes sense for the effective lifetime is the combination of the chemical lifetime and dispersion lifetime (Lu et al., 2015; De Foy et al., 2014). A significant difference is seen in the summer. Our estimation for summer is 1.1 h, while Lange et al. (2022)
reported an ~2 h. With the known observed satellite $NO_2$ line densities, the estimated lifetime will be shorter when the estimated $NO_x$ emissions is higher, and the estimated lifetime shorter otherwise (Jin et al., 2021). Thereby, the reasons to explain the higher summer-to-winter $NO_x$ emission ratio can also be used to explain the shorter summer $NO_x$ lifetime in this study.

### 3.2.3 Interannual variation

Stringent emission control strategies have been implemented in China to control $NO_x$ emissions to combat air pollution since as early as 2010, and a nationwide $NO_x$ emission reduction has been seen since 2012 (Zheng et al., 2018; Li et al.,

2024a). We find a similar trend over Wuhan from 2018 to 2023. Our calculation shows an increase in $NO_x$ emissions over Wuhan from 2018 to 2019 (Fig. 6). The outbreak of COVID-19 in early 2020 led to strong changes in $NO_x$ emissions. Studies have investigated the reduction in satellite $NO_2$ columns (Bauwens et al., 2020; Fioletov et al., 2022) and $NO_x$ emissions (e.g. Ding et al., 2020; Zheng et al., 2021a; Lange et al., 2022; Zhang et al., 2021) around the world due to the lockdown restrictions to combat the COVID-19. Although Wuhan reopened on 9 April 2020, we define the COVID-19 lockdown-influenced period as 23 January to the end of April, considering the slow recovery of human activities. There are 27 days with valid $NO_x$ emissions estimation during this period, as shown in Fig. 6. We calculated the lowest $NO_x$ emissions of less than 20.0 mol s$^{-1}$ during the lockdown, and the mean $NO_x$ emissions in this period are 49% lower than the same period in 2019. The mean emission during the other days in 2020 is comparable to 2019. The EDGAR data does not show decrease in $NO_x$ emissions during the lockdown period, while MEIC reveals ~40% reduction (Fig. 2).

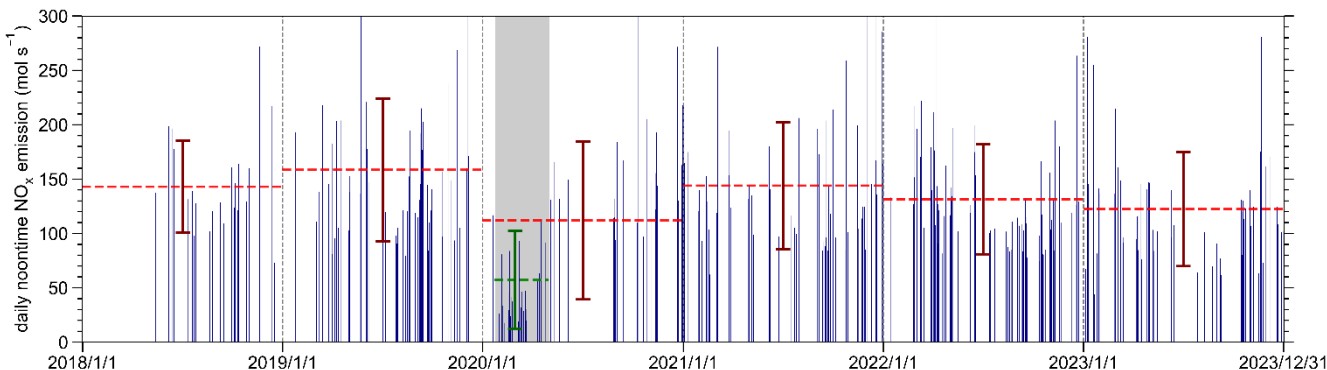

**Figure 6: Daily noontime $NO_x$ emissions over Wuhan on the days with valid estimation from 13 May 2018 through 31 December 2023. The annual mean emissions for each year are marked as red dashed lines. The green dashed line represents the mean $NO_x$ emission during the COVID-19 lockdown-influenced period. The error bars denote the emission standard deviation for each time frame. The COVID-19 lockdown-influenced period is shaded in grey.**

We present the estimated annual mean $NO_x$ emissions in Fig. 6, and find a steady decrease after 2019, except the valley in 2020. Considering the strong seasonality of the estimated $NO_x$ emissions and the availability of valid days for calculation in different years, when we make the comparison between two years, the annual mean emissions are calculated based on the months available for both years (Lonsdale and Sun, 2023). For example, January and February estimations are absent for 2019, June and July are not available for 2020, therefore March to May and August to December monthly values are used to determine the annual mean emissions for the comparison of the two years. We find that the estimated $NO_x$ emissions in 2020 is 10.7% lower than 2019, Lonsdale and Sun (2023) reported only 5% lower. One possible reason for the difference is that they reported a much higher estimation in October 2020. Emissions in 2021 and 2022 are close to each other and about 10% lower than 2019. A significant decrease (13.6%) is seen in 2023 compared to 2022.

### 3.3 Wind field dependence of emission and lifetime estimations

Ideally, $NO_x$ emissions and chemical loss rate directly derived from the satellite observations in combination with the wind fields should be insensitive to the wind direction and even wind speed. However, Valin et al. (2013) showed that the chemical $NO_x$ lifetime in a city plume is wind speed dependent and shorter under strong winds. The EMG method is found to provide best emission estimation under stronger wind speed condition (De Foy et al., 2014; Ialongo et al., 2014). In this study, we also investigate the superposition column model performance on different wind direction and wind speed groups.

### 3.3.1 Wind direction

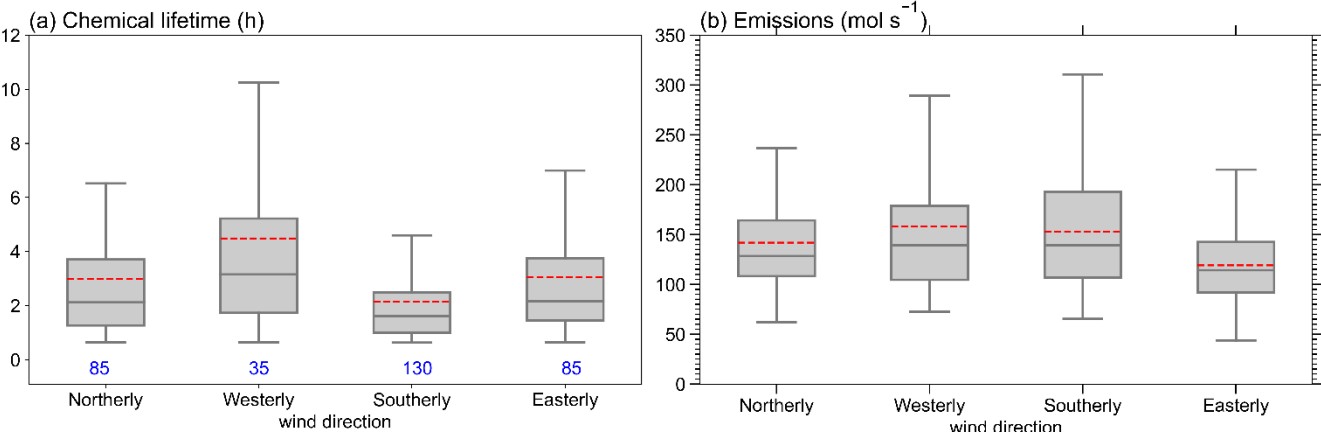

**Figure 7: Boxplots of estimated $NO_x$ (a) chemical lifetimes and (b) emissions over Wuhan are categorized into four groups based on wind direction. For each box, the middle line indicates the median; the box top and bottom indicate the upper and lower quartiles, respectively; the whiskers indicate the farthest nonoutlier values; and the means are presented with red dashed lines. The number of days in each category is listed in blue.**

We separate the 345 valid calculations by wind direction (northerly wind: $315°－45°$, westerly wind: $225°－315°$, southerly wind: $135°－225°$, and easterly wind: $45°－135°$) and compare the estimated $NO_x$ lifetimes and emissions in each category in Fig. 7. The variation in $NO_x$ chemical lifetimes with wind direction is closely related to the seasonal prevailing wind direction. About 1/3 of the days with westerly winds are in the winter and more than half of the westerly wind days are in the cold months; consequently, we compute the longest $NO_x$ lifetimes under westerly winds. On the contrary, spring and summer are dominated by southerly wind, resulting in the shortest $NO_x$ lifetime under this wind direction.

The estimated $NO_x$ emissions also vary with wind directions, as shown in Fig. 7b. Emissions under westerly (southerly) winds are 12% above (8% below) the 345 days' mean emission level, in accordance with the distribution of $NO_x$ chemical lifetimes with the wind direction. Both the northerly and easterly winds are most abundant (~50%) in autumn, but estimated emissions under northerly winds are equal to the 345 days' mean level, while it is ~15% lower under the easterly wind. When looking at tropospheric $NO_2$ columns over Wuhan (Fig. 1a), we find that it is relatively clean to the north, west, and

south of Wuhan, but there are high $NO_2$ spots to the east of the city. Although we have tried to avoid the influence of upwind

emissions by assuming a linear change in the background $NO_2$ columns (Zhang et al., 2023), the finding here indicates that it

is insufficient. The high $NO_2$ columns at the starting point of the $NO_2$ line density will lead to underestimation of city $NO_x$

emissions; the days with distinct upwind $NO_x$ emissions should be treated cautiously in future calculations.

### 3.3.2 Wind speed

We then sort our 345 days with valid $NO_x$ emission calculations into four categories according to wind speed: slow wind

(0–3 m s$^{-1}$), medium slow wind (3–5 m s$^{-1}$), medium fast wind (5–7 m s$^{-1}$), and fast wind (> 7 m s$^{-1}$). We have more than 23

valid calculations for each category, and the corresponding lifetime and $NO_x$ emissions are shown in Fig. 8.

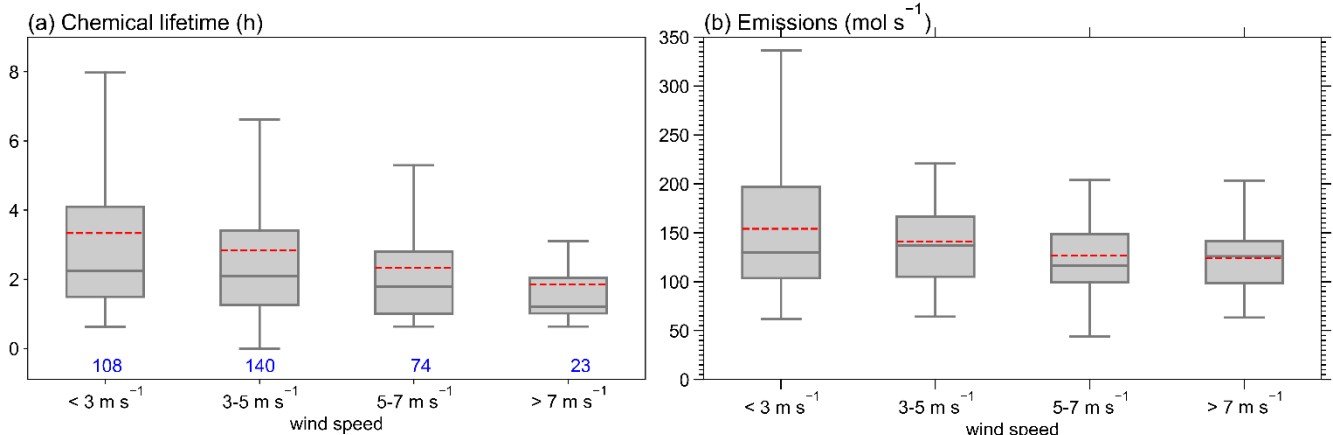

**Figure 8: Same as Fig. 7 but with the wind speed.**

Overall we find a decrease of $NO_x$ chemical lifetime and emission with the increase of the wind speed, which can be

explained that the fast wind speed leads to strong ventilation of $NO_2$. Estimated $NO_x$ chemical lifetime decrease by 14－20%

as the wind speed increases (Fig. 8a). The lifetime under fast wind is nearly 40% lower than that when the wind speed is

slower than 5 m s$^{-1}$, in accordance with the 31% difference calculated by Valin et al. (2013). The change of estimated $NO_x$

emissions with wind speed is smaller (Fig. 8b). The estimated $NO_x$ emission decreases by ~20% from < 3 to 5－7 m s$^{-1}$ wind

speed category, and the emission changes little when wind speed is greater than 5 m s$^{-1}$. Thus the superposition column

underestimate $NO_x$ emissions and chemical lifetimes when the wind speed is faster than 5－7 m s$^{-1}$. The underestimation rate

depends on the fraction of days with fast speed, in Wuhan's case, the overall influence is less than 4% for emission and ~8%

for chemical lifetimes.

Then we look back to the varying estimated $NO_x$ emissions in different seasons and under different wind directions. The

average wind speeds in the four seasons change small from 3.8 to 4.8 m s$^{-1}$, falling into the 3－5 m s$^{-1}$ wind speed group.

This indicates that the lower summer-to-winter $NO_x$ emission ratio estimated by the top-down method (this study and Lange

et al. (2022)) than the bottom-up emission inventories cannot be attributed to the decreasing estimation with the wind speed. In addition, the average wind speeds in the four directions vary from 3.5 to 4.5 m s$^{-1}$, eliminating the influence of the decreasing estimation with the wind speed on the lower estimated emissions under easterly winds.

## 4 Discussion on the uncertainty

We use the superposition column model to estimate city $NO_x$ emissions and chemical lifetimes on daily basis, and the results are used to analyse the temporal variability of $NO_x$ emissions over Wuhan. There are factors leading to uncertainties in our estimation and analysis.

    The uncertainties of $NO_x$ emissions and lifetimes estimation come from the parameters and quantities used during the fitting procedure. The primary sources of uncertainties include:

－The systematic error in TROPOMI $NO_2$ data. The TROPOMI version 2.4.0－2.6.0 data made significant improvements from the earlier versions, and there is a ~20% difference compared to the ground-based data at the Xianghe site in North China (Keppens and Lambert, 2023); we have corrected the possible underestimation over Wuhan by a factor of 1.2, but we thus we still conservatively consider a 20% uncertainty in the $NO_2$ column.

    －Upwind emissions. We find a 15% anomaly in estimated $NO_x$ emission and 8% anomaly in $NO_x$ chemical lifetime for 380 Wuhan when there are distinct hotspots of $NO_2$ in the upwind region. We thus take the 15% (for emission) and 8% (for chemical lifetime) uncertainty to represent the influence of upwind emissions. We must clarify that this part of the uncertainty is 'city-specific'. It can be neglected for some isolated cities or large point sources like Paris or Riyadh. For other cities located in polluted backgrounds, the uncertainty induced by upwind emissions should be calculated accordingly, and it is not necessarily to be 15% and 8%.

－Bottom-up $NO_x$ emissions. We use bottom-up emissions to constrain our estimations. Uncertainty in bottom-up emission inventory is 35% (Zhao et al., 2013; Li et al., 2024b), and we have found that it has a 10% influence on the estimated $NO_x$ emissions while 30% on the chemical lifetime estimation.

    －The $NO_x/NO_2$ ratio. Uncertainty arises from the $NO_x/NO_2$ ratio is 10%, in accordance with that from Zhang et al. (2023).

    －Wind fields. Uncertainty caused by the wind fields partly comes from the systematic wind error, which we take 20% 390 following Zhang et al. (2023); the other part comes from the limitation of the superposition column model, and we exert 4% for emission and 8% for chemical lifetime estimation. This part of uncertainty is also 'city-specific', and the uncertainty increases as the mean wind speed increases in the study domain.

    Finally, we use the root-mean-square sum of all the above contributions, resulting in a 35% uncertainty for the $NO_x$ emissions and 44% for $NO_x$ chemical lifetimes estimated for Wuhan. It is noteworthy that Zhang et al. (2023) considered the 395 uncertainty caused by the area of the study domain and the chemical transport model simulated OH concentration, and they found a 15% and 20% uncertainty caused by the two factors, separately. In this study, we leave out the consideration of the uncertainty caused by the size of the study domain because our study domain is limited to the urban area of Wuhan, and it is

a proper size to estimate NO$_x$ emissions in the urban area. we discard the model simulated OH concentration in the fitting procedure so the uncertainty caused by the OH concentration is avoided. We give initial guess of the NO$_x$ chemical lifetimes, the uncertainty of which is not considered because we let it vary in a quite large range, the initial value would have little influence on the fitting.

Our analysis of the temporal variability of the estimated NO$_x$ chemical lifetime and emission is also of uncertainty, though this part of uncertainty is difficult to quantify. First, our conclusion about the insignificant weekly effect of NO$_x$ emission over Wuhan might be reliable, for this is also found in the surface NO$_2$ and O$_3$ concentration (Wei et al., 2022; Yang et al., 2020), indicating no significant weekday/weekend difference in the oxidation environment. Second, we have overestimated the summer-to-winter emission ratio (0.77), though it is even higher in the bottom-up emission inventory. For one reason, as we have stated in Section 3.2.2, we use a fixed NO$_x$/NO$_2$ ratio, which is found to be lower in summer than winter, so the fixed value will lead to lower estimated summer-to-winter emission ratio. For another, even though the v2.3.1 and after versions have higher retrieval in winter and over polluted area (Van Geffen et al., 2022), they are still found to be lower in polluted area and higher in clean area, when compared to the ground measurements (Keppens and Lambert, 2023). As a consequence, we may still underestimate winter emissions and/or even overestimate summer emissions, thus leading to a higher estimated summer-to-winter emission ratio. Third, when comparing two years' emissions, the annual mean emissions are calculated based on the months available for both years instead of all valid days' average. In this way we have minimized the uncertainty caused by the different availability of valid days in different years. However, the reliability of our results need more years of estimation and more bottom-up information to confirm.

**5 Conclusion**

In this work, we use the superposition column model to calculate city NO$_x$ emissions and lifetimes on a daily basis, with a time span of nearly six years, from May 2018 to December 2023. The city of Wuhan is used as an example to investigate the seasonal pattern, weekly cycle, and interannual variation of city NO$_x$ emissions. The dependence of the model performance on the wind direction and speed are also discussed. We choose the urban area of Wuhan as our study domain, about 45 km radium around the city center. For each full-NO$_2$-coverage day with a homogeneous wind field, the TROPOMI NO$_2$ columns are sampled in to a 15 × 15 grid that is align with the wind direction. Then, every 15 grid cells perpendicular to the wind direction are accumulated to form the NO$_2$ line density. The NO$_2$ line density is fitted with the superposition column model to obtain the final daily NO$_x$ emissions and lifetimes.

For the period from May 2018 to December 2023, we obtain 372 days with valid NO$_x$ emissions and lifetimes estimations over Wuhan, with more than 60 days for each season and each year (2018 excepted) and more than 40 days for each day of the week. The monthly NO$_x$ emissions from 2018 to 2022 are compared with those from EDGAR v8.1 and MEIC v1.4 (until 2020) bottom-up emission inventories. The difference between our estimation and the bottom-up emissions inventories is within 35% for 2018−2019, and EDGAR v8.1 emission is higher by more than 50% than our estimation in 2020−2022. We find a 0.77 summer-to-winter NO$_x$ emission ratio, which might be biased high, though it is even higher in the bottom-up

inventory (> 0.85). The estimated noontime $NO_x$ lifetimes vary from 1.1 h in summer to 4.1 h in winter, with an average of 2.8 h. We see insignificant weekly cycle for Wuhan, which may be related to the living culture and style of the Chinese. We find that $NO_x$ emissions over Wuhan during the COVID-19 lockdown-influenced period are nearly 50% lower than the normal level and rebound to the 2019 emission level for the rest of 2020. Overall, our calculations reveal a steady decline in $NO_x$ emissions from 2019 to 2023.

Compared to previous studies using the superposition column model (Lorente et al., 2019; Zhang et al., 2023), we discard the CTMs simulated OH concentration in the derivation of $NO_x$ chemical loss rate. By doing so we make this method CTM-independent and computational efficient, and also avoid the uncertainty caused by the OH concentration. We use the bottom-up emission inventory to constrain the emission estimation, which induces about 10% uncertainty to the estimation and leads to overestimation of the summer-to-winter emission ratio.

We separate the 345 days (27 days during the COVID-19 lockdown excluded) of $NO_x$ emissions and lifetimes according to wind direction and speed to investigate the model performance under the wind field influence. We find a ~15% lower estimated $NO_x$ emissions in the condition with distinct upwind emissions, indicating that we need to be more careful in the future when computing $NO_x$ emissions over cities or large point sources located in a polluted background. Because of the ventilation of $NO_2$ under fast wind speed, the estimated $NO_x$ chemical lifetime and emission decrease as the wind speed increases. Thereby, the estimation will be underestimated for the sources dominated with fast winds.

We have demonstrated in this work that by combining the superposition column model and the high spatial resolution TROPOMI $NO_2$ column product, one can investigate the variability of $NO_x$ emissions and lifetimes on daily to annual time scale. We also provide recommendations for dealing with conditions with upwind emissions and high wind speeds for better and more accurate city $NO_x$ emissions estimations. So far, the superposition column model has only been used for two cities, Paris and Wuhan; we will extend it to other cities and emission sources in the future.

**Author contributions.** QZ and KFB designed the research. QZ did the processing, visualizations and main writing. KFB edited the paper. CL and AM provided improvements in the method. BZ and SL provided the ABACAS emission inventory. YP reviewed the paper.

**Competing interests.** The authors declare no competing financial interest.

**Acknowledgement.** This work is supported by the National Natural Science Foundation of China (grant no. 42375106 and 41805098) and the National Key R&D Program of China (No. 2023YFB3907500). The Copernicus Sentinel-5P level-2 $NO_2$ data are employed in this work. Sentinel-5 Precursor is a European Space Agency (ESA) mission on behalf of the European Commission (EC). The TROPOMI payload is a joint development by ESA and the Netherlands Space Office. The Sentinel-5 Precursor ground segment development has been funded by ESA and with national contributions from the Netherlands,

Germany, and Belgium. The wind fields used in this study are provided by ECMWF ERA5. EDGAR v8.1 Global Air Pollutant Emissions are provided by https://edgar.jrc.ec.europa.eu/dataset_ap81 (last access: 2 December 2024). We Thank Professor Qiao Ma from Shandong University for performing the comparison between result from this study and the MEIC
v1.4 emission inventory.

**Data availability.** The TROPOMI $NO_2$ data can be freely downloaded from the Tropospheric Emission Monitoring Internet Service (https://www.temis.nl/airpollution/no2.php). The ERA5 data can be found at the Copernicus Climate Change (C3S) climate data store (CDS) (https://cds.climate.copernicus.eu/cdsapp#!/dataset/reanalysis-era5-pressure-levels?tab=overview).

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
