# Peer review of "Estimating the variability of NOx emissions from Wuhan with TROPOMI NO2 data during 2018 to 2023"

_EGUsphere, 2024_

## Referee Comment (RC1)

Review of the manuscript: "Estimating the variability of NOx emissions from Wuhan with TROPOMI NO2 data during 2018 to 2023"

**General comments**
The manuscript employs the superposition column model (previously published in literature) in combination with TROPOMI tropospheric NO2 column data to estimate city-scale NOx emissions and lifetimes and their variabilities. The paper is an extension of a previous work from the same (almost) authors covering a longer period, which allows for the study of the seasonal, weekly and interannual variability. Overall, the manuscript is well written, but in my opinion, there are some parts of the methodology that requires clarification. I suggest publications if the following issues are properly addressed:

**Specific comments**
Methodology (Sect 2.3): The explicit definition of lifetime and of the final emission E appears to be missing.

L164 "The terms $Ei$, $k$, and $\alpha$ are fitted" what about the background coefficient b?

L164- Concerning the OH concentrations, if I understand properly, you use that information to constrain the fitted k coefficient. Is this needed to obtain a "good fit"? Is this worth running a full CTM? What would happen if you let the fit run free (or set a reasonable fixed range), so that you would be not dependent on CTM outputs? What is the variability of the monthly OH? I suppose that if it changes a lot, it makes sense to have a dynamic initial guess, but could you discuss more your choices in this regard? I ask this because, you are making a case for data-driven emission estimation methods, but you still need model data to make your method work. This should be mentioned, I think.

L171 "We restrict the emissions to a gaussian shape" It is not clear how you do that, could you clarify?

L171-172 "a scale factor is applied to the emission term. It is found to be ~0.1 for all the days that lead to the best fit of the NO2 line densities."
It is not clear where this number comes from: what do you mean with "best fit"? Also, does this mean that you are minimizing the difference between your estimates and the inventory? This sounds strange if you then evaluate your estimates against the same inventory. Can you clarify?

L183 "We also exclude the days with estimated NOx emissions beyond 0.5-1.5 times the ABACAS bottom-up emissions." Why do you exactly do that? I read your reasoning concerning the uncertainty and the seasonal variability, but I think you could include also "bad" results as well or at least provide some statistics about them. How many of such days are there? What are the possible reasons for disagreements?

L254 "Their much lower summer-to-winter emission ratio may be caused by much lower

estimated summertime NOx emissions or much higher winter emissions or both." This sentence is maybe a bit self-evident. Are there any specific difference to be mentioned here?

L271-273 "In this work, the a priori NOx emissions are used to restrict the computation of NOx emissions. Thereby, we have partly avoided the possible underestimation of NOx emissions." This is again what might be problematic. If you restrict the computation of the emission to the a priori inventory-based information, is it right to verify your estimates against those same emission inventory values? And, in general, if you need a good bottom-up inventory for your method to perform well, what is the added value of the satellite-based estimates? What would happen without that emission term in the cost function?

L387 "the difference is only 4.7% compared to the ABACAS inventory." Again, the satellite-based emissions are limited to remain close to the ABACAS inventory, so a smaller difference is expected.

Conclusions: you could more thoroughly comment on the limitations of the method, such as the dependence on CTM data and on bottom-up emission inventory data.

**Technical corrections**

Abstract: TROPOMI should be defined

L39 you should probably add a more general (maybe also older) references to this first statement.

L57 It should be noted that the superposition column model presented here is also dependent on CTM (via OH), so it does not solve the issue of running such complex models.

L58 Beirle et al. (2011) actually do not use plume rotation, but they separate the data in 8 classes based on wind direction and then fit the EMG function. Rotation and EMG together were used for example by Lu et al. (2015) among many others. https://acp.copernicus.org/articles/15/10367/2015/

L60 Empirical Modified Gaussian model (EMG) -> this is actually Exponentially-Modified Gaussian model

L62 applied (... -> this is not a complete reference list, add e.g. at the beginning of the references

L91 10-15% there is tilde instead of a dash line here.

L144-145 "rotate the grid map toward the mean wind direction" I would avoid the word rotation here as plume rotation is often used to indicate another method (e.g. Fioletov et

al. 2017). This is actually just a resampling to a grid aligned with the wind direction as you properly described in the caption of Fig. 1.

Fig. 1 panel a: in the title: origional -> original

L114-124
Does it mean that you only directly use GEOS-CHEM data for the initial value of [OH]? Maybe you could clarify this a bit.

L191-192 "There are least valid days in winter (December to February) after spring (March to May) for the cloudy and polluted conditions in winter." not sure what you mean here, could be "There are least valid days in winter (December to February) due to the cloudy and polluted conditions."

L240 To verify this, it would be useful to check some traffic data in the city, if publicly available.

L242 Add references here.

L299 "under 2022" you mean as compared to or lower than 2022?

L344 "It has a small influence (less than 1% in Wuhan's case) on the overall estimation of city NOx emissions, for the days with fast wind make up only less than 10% of the total number of days." The grammar here is a bit off, please rephrase.

L415 "The Wind fields" the world wind should not start with capital letter.

---

## Author Comment (AC1)

Dear Reviewer,

We really appreciate your careful review and insightful comments that helped a lot to improve the analysis and writing of the manuscript. the point-by-point response to your comments is listed below and the revisions/additions/edits are shown in the tracked-change file.

The manuscript presents an interesting investigation using TROPOMI NO2 column data in combined with superposition column model to estimate the emission and lifetime of NOx. Specifically, the study focuses on the derive the NOx emissions and lifetime over Wuhan for 335 clear sky days between May 2018 and December 2023, with the variability of emissions being evaluated to investigate the effectiveness of the emission control strategy. There are some interesting findings resulted from the study. However, the reviewer has some concerns about the novelty of the methods and the significance of the results. See detailed comments below.

**Major comments:**

1. This paper looks like an extension of the authors' ACP paper published in 2023. Similar methods are applied to the TROPOMI data (with version change though) over the same region, and the main difference is that this study extends the study period from 2019-2020 to 2019-2023. Because of the overlap with the authors' previous study, the reviewer is concerned about the novelty of this manuscript, especially since the technical approach has been proposed in their 2023 paper. The authors should clarify in the introduction how this manuscript differs from the previous study, and what would be the novelty of this study.

Response: Thank you for the comment. In the revision of the work, we made substantial modification to the superposition column model. We discard the GEOS-Chem simulated OH concentration in the estimation of $NO_2$ chemical loss rate to get rid of the dependence on CTMs and reduce computational burden, and also avoid the uncertainty induced by the OH concentration. We have added this information in Page 3, Line 75-78 in the revised manuscript. This work is not just an extension in study period of the previous work, we have thoroughly discussed the uncertainty and limitations of the superposition column model on every aspect, providing a reference for future studies to use satellite data to constrain NOx emissions.

2. It's unclear how the NOx lifetime is calculated in GEOS-Chem. The model approach gives an effective lifetime of the entire plume, but the actual chemical lifetime can vary from source to downwind. The effective lifetime can be further confounded by mixing of plumes from multiple directions. I'd suggest the authors clarify the meaning of lifetime in the manuscript, and the limitations of using the model approach to estimate NOx lifetime.

Response: The reviewer's comment is taken and we have made it clear in the revised manuscript that the 'lifetime' mentioned in this work is the 'chemical lifetime' of $NO_x$, and yes the method gives mean chemical life of the entire study domain. The chemical lifetime explains only a part of $NO_x$ loss in the atmosphere, and $NO_x$ chemical lifetime estimated from this method is found to decrease when wind speed increases, which is caused by the stronger ventilation of $NO_x$. We have discussed this in Page 14-15, Line 354-368 in the revised manuscript.

3. The authors showed strong dependence of the emissions and lifetimes on wind field, which does not necessarily mean the NOx emissions vary with wind, but rather due to the limitation of the model and the way the model defines background NO2. This is not a scientific finding, so I think it's better to be included in the uncertainty discussion

Response: We agree with the reviewer's point that the dependency of the estimation on the wind field reveals the uncertainty of the method.

The lower estimation of $NO_x$ emissions under easterly winds indicates that the method underestimates $NO_x$ emissions when there is $NO_2$ hot spots in the upwind region of the study target.

We find that because of the ventilation, the estimated $NO_x$ chemical lifetime and emissions decrease as the wind speed increases. The estimated $NO_x$ emission decreases by ~20% from < 3 to 5－7 m s$^{-1}$ wind speed category, and the emission changes little when wind speed is greater than 5 m s$^{-1}$. Thus the superposition column underestimate $NO_x$ emissions and chemical lifetimes when the wind speed is faster than 5－7 m s$^{-1}$). The underestimation rate depends on the fraction of days with fast speed, in Wuhan's case, the overall influence is less than 4% for emission and ~8% for chemical lifetimes. We have added this discussion in the revised manuscript in Sect. 3.3.2.

4. Figure 4: Please add error bars to this figure to reflect day-to-day variability. Considering the large variability of emissions and the uncertainties of the model and satellite observations, is the weekly cycle statistically significant?

Response: The reviewer's point is well taken and we have added error bars in the day to day variability of $NO_x$ emissions on each day of the week. We agree with the reviewer that there is no significant weekly cycle on $NO_x$ emissions in Wuhan, and the same finding is also found by the surface $NO_2$ and $O_3$ concentration (Wei et al., 2022; Yang et al., 2020) and the traffic flow in Wuhan (https://jtj.wuhan.gov.cn/znjt/zxdt/202409/t20240904_2450210.shtml, last access: 25 November 2024, in Chinese)

5. Section 3.2.3: Considering the large uncertainties of satellite retrievals on daily basis and the potential influences of winds, I think performing the EMG

approach or superposition model over the long-term average data may actually be a better choice for studying the inter-annual variability. I don't see any values added from performing the approach on daily basis. I suggest the authors clarify why it's necessary to calculate daily emissions here.

Response: Thank you for the comment. We agree with the reviewer that there is large uncertainty in satellite retrievals on daily basis, and this is why we did not analyze the variation of $NO_x$ emissions on daily basis, instead we classify the daily emissions into months, seasons, workdays, weekends, and different wind directions and wind speeds. Performing the EMG method over long-term or short term average data is a good choice since there would not be large variation in $NO_x$ emissions during a short time.

However, we argue that it is still necessary to estimate the $NO_x$ emissions on daily basis. First, some unexpected anomaly in $NO_x$ emissions can be identified. For example, Lorente et al. (2019) found highest $NO_x$ emissions on cold weekdays in February 2018 and lowest emissions on warm weekend days in spring 2018, indicating the large contribution from home heating to Paris $NO_x$ emissions. Second, the superposition column model estimate $NO_x$ chemical lifetime and emission through a single overpass of TROPOMI data, avoiding the bias caused by using the averaged $NO_2$ columns in the nonlinear system (Valin et al., 2013). Third, it is more reasonable to use daily $NO_x$ estimation to infer the co-located $CO_2$ emission in the city area, because we need the simultaneous and co-located $NO_2$ and $CO_2$ observation to ensure an accurate estimation (Zhang et al., 2023).

Please refer to Page 2-3, Line 69-75 in the revised manuscript.

**Minor Comments:**

Line 48: Please change "ultraviolet/visible" to "ultraviolet (UV)/visible," and use the acronym "UV" for subsequent mentions throughout the manuscript. (Line 85)

Response: Done.

Figure 1b: Better to show the rotated plume with wind direction as x axis, and cross-wind direction as y axis.

Response: Done. Please refer to the Figure 1b in the revised manuscript.

Line 65: EMG model has been used to estimate episodic fire NOx emissions, which does not need long-term average [1]. The key is to find distinguishable plumes from TROPOMI data.

Response: Thank you for the comment, this sentence has been removed.

Line 136: For the title of Figure 1a, please change "origional" to "original." Additionally, could you indicate the location of Wuhan city on the map and include the corresponding radius value?

Response: Done. Please refer to the Figure 1 in the revised manuscript.

References:

[1] Jin X, Zhu Q, Cohen RC. Direct estimates of biomass burning NOx emissions and lifetimes using daily observations from TROPOMI. *Atmos Chem Phys* 2021; 21: 15569–15587.

References:
Lorente, A., Boersma, K. F., Eskes, H. J., Veefkind, J. P., van Geffen, J., de Zeeuw, M. B., Denier van der Gon, H. A. C., Beirle, S., and Krol, M. C.: Quantification of nitrogen oxides emissions from build-up of pollution over Paris with TROPOMI, Scitific Report, 9, 20033, doi: 10.1038/s41598-019-56428-5, 2019.
Valin, L. C., Russell, A. R., and Cohen, R. C.: Variations of OH radical in an urban plume inferred from $NO_2$ column measurements, Geophysical Research Letters, 40, 1856-1860, 10.1002/grl.50267, 2013.
Wei, J., Liu, S., Li, Z., Liu, C., Qin, K., Liu, X., Pinker, R. T., Dickerson, R. R., Lin, J., Boersma, K. F., Sun, L., Li, R., Xue, W., Cui, Y., Zhang, C., and Wang, J.: Ground-Level $NO_2$ Surveillance from Space Across China for High Resolution Using Interpretable Spatiotemporally Weighted Artificial Intelligence, Environmental Science & Technology, 56, 9988-9998, doi: 10.1021/acs.est.2c03834, 2022.
Yang, G., Liu, Y., and Li, X.: Spatiotemporal distribution of ground-level ozone in China at a city level, Scientific Reports, 10, 10.1038/s41598-020-64111-3, 2020.
Zhang, Q., Boersma, K. F., Zhao, B., Eskes, H., Chen, C., Zheng, H., and Zhang, X.: Quantifying daily $NO_x$ and $CO_2$ emissions from Wuhan using satellite observations from TROPOMI and OCO-2, Atmospheric Chemistry and Physics, 23, 551-563, doi: 10.5194/acp-23-551-2023, 2023.

---

## Author Response (AR1)

Dear Editor,

We really appreciate your and the three reviewers' efforts and insightful comments to improve the analysis and writing of the manuscript. the point-by-point response to the comments is listed below and the revisions/additions/edits are shown in the tracked-change file.

**Reviewer #1:**

Review of the manuscript: "Estimating the variability of NOx emissions from Wuhan with TROPOMI NO₂ data during 2018 to 2023"

**General comments**

The manuscript employs the superposition column model (previously published in literature) in combination with TROPOMI tropospheric NO2 column data to estimate city-scale NOx emissions and lifetimes and their variabilities. The paper is an extension of a previous work from the same (almost) authors covering a longer period, which allows for the study of the seasonal, weekly and interannual variability. Overall, the manuscript is well written, but in my opinion, there are some parts of the methodology that requires clarification. I suggest publications if the following issues are properly addressed:

**Specific comments**

Methodology (Sect 2.3): The explicit definition of lifetime and of the final emission E appears to be missing.

Response: Thank you for the comment. In this study, $NO_x$ 'lifetime' is the 'chemical lifetime', and we have clarified this in the revised manuscript Sect. 2.4: 'In Eq. (1), $k$ (s⁻¹) represents the loss rate of $NO_2$ at the TROPOMI overpass time, and the relationship between $k$ and $NO_2$ chemical lifetime $\tau_{[NO_2]}$ (h) is $k = \frac{1}{\tau_{[NO_2]} * 3600}$', and the $NO_x$ chemical life is determined by $\tau_{[NO_x]} = \tau_{[NO_2]} \cdot \frac{[NO_x]}{[NO_2]}$. We have also changed the term 'lifetime' to 'chemical lifetime' throughout the revised manuscript.

The definition of the final emission E is defined as 'the total $NO_x$ emissions $E$ (in the unit of mol s⁻¹) from the study domain can be calculated with $E = \sum_{i=1}^{15} E_i \times L$'.

Please refer to Page 6-7, Line 163-164 and Line 180-183 in the revised manuscript.

L164 "The terms $Ei$, $k$, and $\alpha$ are fitted" what about the background coefficient b?

Response: $b$ is also fitted. The sentence has been rephrased to: 'The terms $E_i$, $k$, $\alpha$ and $b$ are fitted through a least squares minimization to the TROPOMI observed $NO_2$ line densities ($N_{TROPOMI}(x)$) and the a priori ABACAS $NO_x$ emissions ($E_{ABACAS,i}$) to determine $N(x)$.' Please refer to Page 6, Line 173-174 in the revised manuscript.

L164- Concerning the OH concentrations, if I understand properly, you use that

information to constrain the fitted k coefficient. Is this needed to obtain a "good fit"? Is this worth running a full CTM? What would happen if you let the fit run free (or set a reasonable fixed range), so that you would be not dependent on CTM outputs? What is the variability of the monthly OH? I suppose that if it changes a lot, it makes sense to have a dynamic initial guess, but could you discuss more your choices in this regard? I ask this because, you are making a case for data-driven emission estimation methods, but you still need model data to make your method work. This should be mentioned, I think.

Response: Thank you for the insightful comments. Yes, at first we needed the CTM output OH concentration to constrain the fitted k coefficient, and we ran full CTM for the monthly OH. The OH concentration changes a lot from month to month, the summer value can be several times higher than the winter value. We agree that this contradicts with the 'CTM-independent method' that we have claimed in the introduction. In the revision of the work, the CTM output OH concentration is no longer used in the fitting of the $NO_2$ line density to make this method free from the CTMs.

Instead, we use an initial guess of the $NO_2$ chemical lifetime of 4 h for the cold months (October to March) and 2 h for the warm months (April to September), the k ($s^{-1}$) coefficient is derived through $k = \frac{1}{\tau_{[NO_2]}*3600}$. During the fitting procedure, we let k changes between 1/4 and 4 times of the initial value. We could not let the fit run free with k, in which case the final $NO_x$ emissions and chemical lifetimes would be determined by the k value, and might result in a very long chemical lifetime in summer or very short chemical lifetime in winter. On the other hand, we could not set the k range too narrow on the condition that k may change significantly from day to day and month to month.

Since that we set a wide range for k, the emission term is used in the cost function to reduce the dependence of the fitted $NO_x$ chemical lifetimes and emissions on k, and we have evaluated that the uncertainty of the a priori emission (35%) can result in 10% and 30% influence on the final estimated emission and chemical lifetime, respectively.

L171 "We restrict the emissions to a gaussian shape" It is not clear how you do that, could you clarify?

Response: Yes. We assume that the $NO_x$ emissions intensities in the city are distributed as a Gaussian shape along with the wind direction: $E_i = amp * \exp[-(x_i - cen)^2/wid]$, the parameter $cen$ is initially set as the location of the city center, and we let it shift along the wind during the fitting procedure, $amp$ and $wid$ are also fitted to obtain $E_i$.

L171-172 "a scale factor is applied to the emission term. It is found to be ~0.1 for all the days that lead to the best fit of the $NO_2$ line densities." It is not clear where this

number comes from: what do you mean with "best fit"? Also, does this mean that you are minimizing the difference between your estimates and the inventory? This sounds strange if you then evaluate your estimates against the same inventory. Can you clarify?

Response: The cost function is expressed as: $func = (\frac{N(x)-N_{TROPOMI}(x)}{N_{TROPOMI}(x)})^2 + fac * (\frac{E_i-E_{ABACAS,i}}{E_{ABACAS,i}})^2$, it is to minimize the difference between the fitted and TROPOMI $NO_2$ line densities and between the estimated emission and the inventory. Here we set a scale factor for the emission term, and the factor is limited to be between 0.1 and 0.2 to make sure that the cost function is dominated by the minimization of the difference between the fitted and TROPOMI $NO_2$ line densities.

The reason we add the emission term in the cost function is because we set a very wide range for the $NO_x$ loss rate k by allowing it change between 1/4 and 4 times of its initial value. Without the constraint from the a priori emission, the final results of the model would be determined by the k value. On the other hand, we could not narrow down the varying range for k, for we have no more accurate information of it.

We agree with the reviewer that it is not appropriate to validate the estimated $NO_x$ emissions with the ABACAS $NO_x$ inventory that is used to constrain the fitting. In the revision, we use the EDGAR v8.1 monthly $NO_x$ emission from 2018 to 2022 and MEIC v1.4 monthly $NO_x$ emission from 2018 to 2020 to validate our estimation. Please refer to the revised Figure 2 and section 3.1 in Page 8 Line 207-222 in the revised manuscript.

L183 "We also exclude the days with estimated NOx emissions beyond 0.5-1.5 times the ABACAS bottom-up emissions." Why do you exactly do that? I read your reasoning concerning the uncertainty and the seasonal variability, but I think you could include also "bad" results as well or at least provide some statistics about them. How many of such days are there? What are the possible reasons for disagreements?

Response: Thank you for the constructive comments. The day to day, monthly and year to year variation and the uncertainty of the $NO_x$ emissions were all within the ±50% range, so the estimated data beyond this range were excluded. The extremely high and low estimation is mainly caused by the bias of the satellite $NO_2$ observation. Overall we obtained 24 days higher than 1.5 times of the a priori emission and 15 days lower than 50% of the a priori emission, the total of them made up about 10% of the total number of the estimations.

Most of the high emission days are in winter and low emission days are in summer, so the filter of them dampened the seasonal variability. For this reason, in the revision of the work, we choose to keep these results for analysis.

L254 "Their much lower summer-to-winter emission ratio may be caused by much

lower estimated summertime NOx emissions or much higher winter emissions or both." This sentence is maybe a bit self-evident. Are there any specific difference to be mentioned here?

Response: Thank you for the comment. We have added discussions on the estimated summer-to-winter $NO_x$ emission ratio in Page 11, Line 276-280 in the revised manuscript:
Two possible factors may contribute to the large difference in the summer-to-winter emission ratio between this study and Lange et al. (2022). First is the different treatment to the $NO_x$-to-$NO_2$ ratio. We use a fixed $NO_x$-to-$NO_2$ ratio of 1.26, while Lange et al. (2022) calculated the ratio from day to day, and it was lower in summer than in winter, leading to a lower $NO_x$ emission estimation in summer. Second is that we use the bottom-up emission inventory to constrain our estimation, the flat seasonality of the bottom-up emissions leads to a higher summer-to-winter ratio of this study.

L271-273 "In this work, the a priori NOx emissions are used to restrict the computation of NOx emissions. Thereby, we have partly avoided the possible underestimation of NOx emissions." This is again what might be problematic. If you restrict the computation of the emission to the a priori inventory-based information, is it right to verify your estimates against those same emission inventory values? And, in general, if you need a good bottom-up inventory for your method to perform well, what is the added value of the satellite-based estimates? What would happen without that emission term in the cost function?

Response: We agree with the reviewer that it is not appropriate to validate the estimated $NO_x$ emissions with the bottom-up emission inventory that is used to constrain the estimation, and in the revised manuscript, we have used the monthly EDGAR v8.1(2018-2022) and MEIC v1.4 (2018-2020) to validate the estimation, please refer to Figure 2, Page 8, Line 204-219 in the revised manuscript.

According to our cost function, the bottom-up emission plays only a small role in it since we give it a scale factor varying from 0.1 to 0.2 in the fitting, the final result is dominated by the minimization between the fitted and TROPOMI observed $NO_2$ line density.

Because of limited information on the $NO_2$ chemical loss rate, we gave a quite large changing range for it, without the constrain from the bottom-up emission inventory, the model will randomly settle down with a loss rate that lead to the minimum difference between fitted and TROPOMI observed $NO_2$ line density. The fitted chemical loss rate would be very high in winter or very low in summer, so we need the emission term in the cost function to keep the loss rate falls into a reasonable range. In the future, if we find a solution to narrow down the changing range of $NO_2$ chemical loss rate during our fitting, the emission term can be removed from the cost function.

L387 "the difference is only 4.7% compared to the ABACAS inventory." Again, the

satellite-based emissions are limited to remain close to the ABACAS inventory, so a smaller difference is expected.

Response: We agree with the reviewer that we should not validate our results with the ABACAS inventory, so we collected the monthly emissions of EDGAR v8.1 from 2018 to 2022 and MEIC v1.4 from 2018 to 2020 to validate the monthly $NO_x$ emission of this work.

[Figure]

Overall, the TROPOMI estimation is close to the bottom-up emission inventories during cold months, while much lower during warm months. For the three years (2018 to 2020) when MEIC v1.4 data is available, the difference between TROPOMI and MEIC v1.4 is within 35%, and both of them capture the $NO_x$ emission reduction in early 2020 due to COVID-19 lockdown. TROPOMI and EDGAR v8.1 are close to each other (within 30% difference) in 2018 and 2019, but the discrepancy is larger since 2020. EDGAR v8.1 is > 50% higher than TROPOMI from 2020 to 2022. Please refer to Figure 2 and Page 8, line 207-222 in the revised manuscript.

Conclusions: you could more thoroughly comment on the limitations of the method, such as the dependence on CTM data and on bottom-up emission inventory data.

Response: The reviewer's suggestion is well taken and we have thoroughly discussed the uncertainty and limitation of this study. Please refer to Sect. 4 in Page 15-16 and Line 436-446 in Page 17 in the revised manuscript.

Technical corrections
Abstract: TROPOMI should be defined
Response: Done.

L39 you should probably add a more general (maybe also older) references to this first statement.
Response: the references here are changed to:
Bassett, M. and Seinfeld, J. H.: Atmospheric equilibrium model of sulfate and nitrate aerosols, Atmospheric Environment, 17, 2237-2252, https://doi.org/10.1016/0004-6981(83)90221-4, 1983.

Penner, J. E., Atherton, C. S., Dignon, J., Chan, S. J., and Walton, J. J.: Tropospheric nitrogen: A three-dimensional study of sources, distributions, and deposition, Journal of Geophysical Research, 96, 959-990, https://doi.org/10.1029/90JD02228, 1991.

Jacob, D.: Introduction to Atmospheric Chemistry, Princeton Univ. Press, 1999.

L57 It should be noted that the superposition column model presented here is also dependent on CTM (via OH), so it does not solve the issue of running such complex models.

Response: Agree. In the revision of the work, we use an initial guess of the $NO_2$ chemical lifetime of 4 h for the cold months (October to March) and 2 h for the warm months (April to September), the k ($s^{-1}$) coefficient is derived through $k = \frac{1}{\tau_{[NO_2]}*3600}$.

In this way we avoid the running of CTM and make this method computational efficient.

L58 Beirle et al. (2011) actually do not use plume rotation, but they separate the data in 8 classes based on wind direction and then fit the EMG function. Rotation and EMG together were used for example by Lu et al. (2015) among many others. https://acp.copernicus.org/articles/15/10367/2015/

Response: The reviewer's point is well taken and the sentence has been rephrased as 'Beirle et al. (2011) reduced the 2-dimensional $NO_2$ map surrounding a large point source (such as a megacity or a power plant, factory) to the 1D $NO_2$ line density by integrating the $NO_2$ column density across the wind direction.' Please refer to Page 2, Line 58-59 in the revised manuscript.

L60 Empirical Modified Gaussian model (EMG) -> this is actually Exponentially-Modified Gaussian model

Response: Corrected.

L62 applied (… -> this is not a complete reference list, add e.g. at the beginning of the references

Response: Added.

L91 10-15% there is tilde instead of a dash line here.

Response: We have changed this expression to '10%－15%'.

L144-145 "rotate the grid map toward the mean wind direction" I would avoid the word rotation here as plume rotation is often used to indicate another method (e.g. Fioletov et al. 2017). This is actually just a resampling to a grid aligned with the wind direction as you properly described in the caption of Fig. 1.

Response: The reviewer's comment is well taken and we have rephrased this sentence

as 'We construct a 15 × 15 grid map centered at the city center with each grid size of 0.05 °× 0.05 °(6km × 6km) toward the mean wind direction. One demission of the grid map along with and the other perpendicular to the wind direction.' Please refer to Page 5, Line 150-152 in the revised manuscript.

Fig. 1 panel a: in the title: origional -> original

Response: Corrected.

L114-124 Does it mean that you only directly use GEOS-CHEM data for the initial value of [OH]? Maybe you could clarify this a bit.

Response: The GEOS-chem simulated OH concentration is no longer used in the work. Instead, we use an initial guess of the $NO_2$ chemical lifetime of 4 h for the cold months (October to March) and 2 h for the warm months (April to September) to derive the k (s$^{-1}$) coefficient: $k = \frac{1}{\tau_{[NO_2]}*3600}$.

L191-192 "There are least valid days in winter (December to February) after spring (March to May) for the cloudy and polluted conditions in winter." not sure what you mean here, could be "There are least valid days in winter (December to February) due to the cloudy and polluted conditions."

Response: Thank you for the comment, this sentence has been changed to: "There are least valid days in winter (December to February) due to the cloudy and polluted conditions.".

L240 To verify this, it would be useful to check some traffic data in the city, if publicly available.

Response: The reviewer's suggestion is well taken and we've found evidence from the 'Annual Report on Wuhan Transportation Development (2023)' (https://jtj.wuhan.gov.cn/znjt/zxdt/202409/t20240904_2450210.shtml, last access: 25 November 2024, in Chinese) that the traffic flow passed through the Outer Ring Road and the Fourth Ring Road of Wuhan was highest on Friday and lowest on Tuesday and Sunday, but the difference is only less than 2%.

This can confirm our finding that there's no significant weekday/weekend difference in $NO_x$ emissions from Wuhan. Please refer to Page 10, Line 252-255 in the revised manuscript.

L242 Add references here.

Response: Added.

L299 "under 2022" you mean as compared to or lower than 2022?

Response: Yes, it should be 'compared to 2022' and it is corrected.

L344 "It has a small influence (less than 1% in Wuhan's case) on the overall estimation of city NOx emissions, for the days with fast wind make up only less than 10% of the total number of days." The grammar here is a bit off, please rephrase.

Response: Thank you for the comment. This sentence is removed in the revised manuscript.

L415 "The Wind fields" the world wind should not start with capital letter.

Response: Corrected.


**Minor Comments:**

Line 48: Please change "ultraviolet/visible" to "ultraviolet (UV)/visible," and use the acronym "UV" for subsequent mentions throughout the manuscript. (Line 85)

Response: Done.

Figure 1b: Better to show the rotated plume with wind direction as x axis, and cross-wind direction as y axis.

Response: Done. Please refer to the Figure 1b in the revised manuscript.

Line 65: EMG model has been used to estimate episodic fire NOx emissions, which does not need long-term average [1]. The key is to find distinguishable plumes from TROPOMI data.

Response: Thank you for the comment, this sentence has been removed.

Line 136: For the title of Figure 1a, please change "origional" to "original." Additionally, could you indicate the location of Wuhan city on the map and include the corresponding radius value?

Response: Done. Please refer to the Figure 1 in the revised manuscript.

Response: Thank you for the comments and we have reorganized the methodology to make it clearer to follow, some details are added.

We construct a 15 x 15 grid map centered at the city center (114.6 °E, 30.6 °N) with each grid size of 0.05 °x 0.05 °(6km x 6km) toward the mean wind direction, with one demission of the grid map along with and the other perpendicular to the wind direction. The mean wind direction is determined by the mean meridional and zonal winds over the study domain. The original TROPOMI observation (Figure 1a) is sampled into the rotated grid (Fig. 1b). The TROPOMI $NO_2$ columns in the 15 grid cells perpendicular to the wind direction are integrated to form the so-called $NO_2$ 'line densities' (Beirle et al., 2011), resulting in 15 grid cells along the wind direction (Fig. 1c).

We treat each cell along with the wind direction with the simple column model proposed by Jacob (1999), finally $NO_x$ emissions from each cell (Ei) are added up to obtain the total emission E in the study domain. The 'chemical lifetime' ($\tau$, h), since its relationship with the chemical loss rate of $k$ ($s^{-1}$) in the superposition column model is $k = \frac{1}{\tau*3600}$.

We narrow down our study domain from the administrative area of Wuhan in Zhang et al. (2023) to the urban area (within the Fourth Ring Road of Wuhan, ~90 km in diameter) in this study considering that, first, most (more than 60%) of the $NO_x$ emissions are concentrated in the urban area; second, we use regional mean wind fields and $NO_x$ chemical loss rate, the larger study domain would induce large uncertainty in the result.

Please refer to the Section 2.4 in the revised manuscript.

I am concerned that some decisions influence the emission estimates, especially the investigated seasonal patterns (see also comments in the specific comment part below):

How does the ABACAS emission inventory filter influence the results? Is the computation of NOx emissions, which seems to be restricted by a priori emissions, dampening the estimated emissions? How much is the method depending on the emission inventory data?

Response: Thank you for the comments. At first, we filter out the estimated $NO_x$ emissions that beyond the $\pm50\%$ around the ABACAS emissions, because the day to day, monthly and year to year variation and the uncertainty of the $NO_x$ emissions were all within the $\pm50\%$ range, the estimation that beyond this range can be seen as anomaly. The extremely high and low estimation is mainly caused by the bias of the satellite $NO_2$ observation.

Overall we obtained 24 days higher than 1.5 times of the a priori emission and 15 days lower than 50% of the a priori emission, the total of them made up about 10% of the total number of the estimations.

Most of the high emission days are in winter and low emission days are in summer, the filter of them dampened the seasonal variability. Therefore, in the revision of the work, we choose to keep these results for analysis.

Is the bias correction factor of 1.2 valid for all seasons?

Response: We use the factor 1.2 for all the days. The TROPOMI data version we used is the version 2.4.1－2.6.1, and even though the v2.3.1 and after versions have higher retrieval in winter and over polluted area (Van Geffen et al., 2022), they are still found to be lower in polluted area and higher in clean area, when compared to the ground measurements (Keppens and Lambert, 2023). As a consequence, we may still underestimate winter emissions and/or even overestimate summer emissions, thus leading to a higher estimated summer-to-winter emission ratio. We have added this discussion in Page 16, Line 405-412 in the revised manuscript.

How would seasonal patterns change using a daily NOx/NO2 ratio instead of a fixed value of 1.26? The loss rate of NOx (k in Eq 1) is based on the fixed NOx/NO2 ratio and does not consider the temperature dependency of the NO2 and OH reaction.

Response: Thank you for this comment. A fixed $NO_x/NO_2$ ratio is adapted in this study, while it is found to be varying by 10% from the lowest to the highest month. As a result, we overestimate the summer-to-winter emission ratio. We have clarified this in Page 11, Line 277-279 in the revised manuscript.

I suggest publication if the raised issues are addressed.

**Specific comments:**

L39: You provide Goldberg et al. (2019) and Zhang et al. (2021) as references for key information about nitrogen oxides. I think it is more appropriate to cite the references which are cited in these references, e.g., Jacob, D. J. Introduction to Atmospheric Chemistry; Princeton University Press, 1999.

Response: Thank you for the comment. We have changed the citation here to:
Bassett, M. and Seinfeld, J. H.: Atmospheric equilibrium model of sulfate and nitrate aerosols, Atmospheric Environment, 17, 2237-2252, https://doi.org/10.1016/0004-6981(83)90221-4, 1983.
Penner, J. E., Atherton, C. S., Dignon, J., Chan, S. J., and Walton, J. J.: Tropospheric nitrogen: A three-dimensional study of sources, distributions, and deposition, Journal of Geophysical Research, 96, 959-990, https://doi.org/10.1029/90JD02228, 1991.
Jacob, D.: Introduction to Atmospheric Chemistry, Princeton Univ. Press, 1999.

L51, L53, L56: You already provide several references for the different emission estimation methods and applications; since these are, however, only some examples, I would change these parts to (e.g., reference1, reference2, …)

Response: The 'e.g.,' has been added to each reference list in Page 2, Line 52-57 in the revised manuscript:
With the improving capabilities of later satellite sensors, more researchers started to estimate $NO_x$ emissions on higher spatial and temporal resolutions but still depended on CTMs (e.g., Ding et al., 2017; Visser et al., 2019; Xing et al., 2022). However, there are barriers to access and employment of CTMs, and there is a substantial computational burden when our target is a single city. Therefore, CTM-independent methods have been developed and applied to estimate $NO_x$ emissions since the early 2010s (e.g., Beirle et al., 2011; De Foy et al., 2014; Kong et al., 2019; 2019; Lorente et al., 2019; Rey-Pommier et al., 2022).

L58: Beirle et al. (2011) have not rotated the NO2 maps, they divided the OMI data into wind sectors based on the present wind direction of the individual measurement and estimated emissions for the eight defined wind sectors. Rotation was first

introduced by Pommier et al. (2013) and Valin et al. (2013), which you mention also in L62.

Response: The reviewer's comment is taken and this sentence has been rephrased as: 'Beirle et al. (2011) reduced the 2-dimensional $NO_2$ map surrounding a large point source (such as a megacity or a power plant, factory) to the 1D $NO_2$ line density by integrating the $NO_2$ column density across the wind direction.' Please refer to Page 2, Line 58-59 in the revised manuscript.

L60-61: Since Beirle et al. (2011) have not only estimated emissions for cities, I would suggest deleting the word city in "city NOx emissions" and replacing city with source in "over the city and its decay downwind of the city."

Response: Done.

L65-66: "relatively large study area", I understand that this is meant probably in comparison to the method you use with a diameter of 90 km, still the EMG method is already possible for individual cities and power plants. Maybe you can clarify this to avoid confusion.

Response: We were trying to say that the EMG model asks the cities and power plants to be isolated from other sources and requires a large fit range. We find that it is not necessary to stress it here so we decide to delete this information.

You wrote that the EMG model is limited to calculating emissions only for data averaged over a longer time period. However, this is more related to the quality of the satellite product used, e.g. Goldberg et al. (2019) showed that using the EMG method with TROPOMI NO2 observations also single overpasses can deliver valuable results, which you are also mentioning in the next paragraph. You can change the sentence to something like "…and with OMI data it is limited to calculating mean NOx emissions from observations over longer time periods, like some years.".

Response: Thank you for the comment. The sentence is rephrased as: 'This method is more frequently used to calculate mean $NO_x$ emissions over longer time periods (like some years) with OMI data (e.g., Lu et al., 2015; Liu et al., 2016).' Please refer to Page 2, Line 64-66 in the revised manuscript.

L69: "Lorente et al. (2019) narrowed down the study area to the domain of one city". Emission estimates for individual cities were already possible with OMI data and the EMG method.

Response: Agree. We have deleted this phrase in the revised manuscript Page 2, Line 69-70: 'Based on a single TROPOMI overpass, Lorente et al. (2019) developed a superposition column model to fit the $NO_2$ line density for daily $NO_x$ emissions'.

L73: Is Valin et al. (2013) here the right reference, I think the superposition column model is not discussed in Valin et al. (2013), shouldn't it be Lorente et al. (2019) or maybe both?

Response: Valin et al. (2013) pointed out that using the average of $NO_2$ concentration to calculate $NO_x$ chemical lifetimes and emissions might lead to some bias because this is a nonlinear system. The superposition column model has avoided this bias by calculating $NO_x$ chemical lifetimes and emissions on daily basis. We have reorganized this sentence to avoid misunderstanding. Please refer to Page 3, Line 73-75 in the revised manuscript.

L87: "unprecedented nadir spatial resolution" Since the resolution of TEMPO with 2km x 4.5km is better than TROPOMI's resolution, I suggest deleting "unprecedented"

Response: Done.

L87: Since you also use data before 6 August 2019, briefly mention the spatial resolution before the change.

Response: Added.

L90-92: You wrote that "The version 2.3.1 includes a different treatment of the surface albedo compared to earlier versions, which led to a 10~15>% increase of tropospheric NO2 columns over polluted scenes." This is misleading, the surface albedo in the NO2 window (OMI LER) and for the cloud product (GOME-2 LER) is replaced with the TROPOMI DLER in v2.4, which you also describe in the following sentences. The main changes in v2.3.1 compared to v1.x are the switch to the FRESCO-wide cloud product and a correction of the surface albedo for cloud-free scenes (only for specific scenes with cloud fractions < 0 and > 1). All changes together result in tropospheric NO2 VCDs that are 10-40% larger than in v1.x.

Response: The reviewer's comment is taken and we have rewritten the description about TROPOMI version 2.3.1: 'Compared to the previous versions v1.x, the version 2.3.1 includes a different treatment of the surface albedo to avoid negative and > 1 cloud fractions, and updates the FRESCO-wide cloud retrieval that leads to a lowering cloud pressure. These result in 10%－40% increase of tropospheric $NO_2$ columns, depending on the level of pollution and season (Van Geffen et al., 2022).' Please refer to Page 3-4, Line 99-101 in the revised manuscript.

L105: You use a scale factor of 1.2 to correct the low bias of the TROPOMI data based on the ground-based validation with the Xianghe station. Can you comment on possible differences (bias variations) between different seasons, might this influence your seasonal investigations?

Response: The reviewer's comment is well taken and we have discussed the fixed scale factor of 1.2 on the seasonal pattern investigation of this study. We use the scale factor 1.2 for all seasons and pollution levels to partly correct the potential low bias of the TROPOMI data. However, even though the v2.3.1 and after versions have higher retrieval in winter and over polluted area (Van Geffen et al., 2022), they are still found to be lower in polluted area and higher in clean area, when compared to the ground measurements (Keppens and Lambert, 2023). As a consequence, we may still underestimate winter emissions and/or even overestimate summer emissions, thus leading to a higher estimated summer-to-winter emission ratio. Please refer to Page 16, Line 408-411 in the revised manuscript.

L112: Why have you decided on the 950hPa level?

Response: Considering the vertical consistency of wind speeds and directions, we use the 3 levels mean meridional and zonal wind below 950hPa. We have clarified this in Page 4, Line 122-123 in the revised manuscript.

L122: Do you use daily OH concentrations, please clarify.

Response: We did not use the daily OH concentration but the monthly mean value in the initial work. During the revision of the work, we discard the OH concentration and chemical loss rate constant k' from the GEOS-chem model in the fitting. Instead, we use 4 hours for the cold months (October to March) and 2 hours for the warm months (April to September) as the initial guess of the chemical lifetime of $NO_2$, which avoids the uncertainty caused by the model output OH concentration and $NO_x$ chemical loss rate and make this method computational efficient.

L124: You decided to use a fixed value of 1.26 for the NOx/NO2 ratio. You mention that it varies less than 10% in season. Can you show the seasonal variation of the NOx/NO2 ratio over the year? I think this is especially relevant as you are investigating seasonal patterns.

Response: Thank you for the comment. The figure below shows the monthly $NO_x/NO_2$ ratio simulated by GEOS-Chem, it is highest in winter and lowest in summer, with the difference of 10.2% (highest/lowest -1). We keep the $NO_x/NO_2$ ratio fixed for all the days, and it leads to high estimation in summer and lower estimation in winter, and thus we overestimate the summer-to-winter emission ratio. We have added this information in the comparison between this work and Lange et al. (2022). Please refer to Page 11, Line 276-280 in the revised manuscript.

[Figure]

L128: Information missing about the ABACAS emission inventory. Is it annual or monthly data, based on which year?

Response: Thank you for the comment. The ABACAS $NO_x$ emission inventory is monthly data for the year 2019. We have added this information in Page 4, Line 126-130 in the revised manuscript.

L130 & Fig. 2: EDGAR provides not only annual but also monthly time series for 2018 NOx, did you have a look at these? Relevant for L212.

Response: The reviewer's suggestion is well taken and we have used EDGAR v8.1 monthly $NO_x$ emissions from 2018 to 2022 and MEIC v1.4 monthly emissions for 2018-2020 to compare with the result from this study. We found that the TROPOMI estimation is close to the bottom-up emission inventories during cold months, while much lower during warm months. For the three years (2018 to 2020) when MEIC v1.4 data is available, the difference between TROPOMI and MEIC v1.4 is within 35%, and both of them capture the $NO_x$ emission reduction in early 2020 due to COVID-19 lockdown. TROPOMI and EDGAR v8.1 are close to each other (within 30% difference) in 2018 and 2019, but the discrepancy is larger since 2020. EDGAR v8.1 is > 50% higher than TROPOMI from 2020 to 2022. Please refer to Page 8, Line 213-222 and the new Figure 2 in the revised manuscript.

Section 2.4: You write you rotate with the mean wind direction. How is the mean wind calculated, over which area? What is rotated? Is there a difference between the wind errors in Fig 1(a) and (b)? How do you determine the diameter of your circle?

Response: Thank you for the comment. We construct a 15 × 15 grid map centered at the city center with each grid size of 0.05 °× 0.05 °(6km × 6km) toward the mean wind direction, with one demission of the grid map along with and the other perpendicular to the wind direction. The mean wind direction is determined by the mean meridional and zonal winds over the study domain. The only difference between Figure 1a and b is that Figure 1a is plotted directly from the TROPOMI observation, and Figure 1b is the rotated result of the study domain. The circle covers the region within the Fourth Ring

Road of Wuhan, and it is the urban area, with a diameter of about 90 km. Please refer to the revised Figure 1 and Line 149-152 in Page 5 of the revised manuscript.

Equation 1: Use points for multiplication signs instead of crosses.

Response: Corrected.

L154-157: k represents the loss rate of NOx. You use a fixed value for the rate constant k' between NO2 and OH. How large is the seasonality due to the temperature dependency of this reaction? How large is the seasonal variation of the NOx/NO2 ratio over the year (see comment above)? Relevant for section 3.2.2 about seasonal patterns:

Response: We thank the reviewer for this comment.

According to Burkholder et al. (2020), the $NO_2$ chemical loss rate constant increases with temperature: $k'(T) = 2.8 \times 10^{-11} \times \frac{T}{300}$. In Wuhan's case, the seasonal variation of $k'$ would be ~8% (the temperature record shows that in 2023 the lowest monthly mean noon temperature is in February of 11°C,and the highest is 34°C in July); Based on the GEOS-Chem model simulation, the seasonal variation of $NO_x/NO_2$ ratio is ~10%.

In our initial work, both the fixed k' and $NO_x/NO_2$ ratio lead to overestimation of the summer-to-winter emission ratio. In the revision of the work, we discard the OH concentration from GEO-Chem model, instead we give an initial guess for $NO_2$ lifetime τ (h) of 2 h in warm months (April to September) and 4 h in cold months (October to March) and the $NO_x$ chemical loss rate k (s$^{-1}$) is derived by $k = \frac{1}{\tau * 3600}$ · $\frac{[NO_2]}{[NO_x]}$.

The fixed $NO_x/NO_2$ value will dampen the seasonality of $NO_x$ emission estimation, and we have added this discussion in Page 11, Line 276-280 in the revised manuscript.

Equation 3: for x >xi (?) missing

Response: It is not for x > xi, it is for all the x. We have placed equations 1-3 together to make it clearer. Please refer to Page 6, Line 159-161 in the revised manuscript.

L163: What do you mean with "upend point", at the upwind end point of the grid/city/circle?

Response: Yes, it is the upwind end point of the city. It is corrected in the revised manuscript.

L166: You write you use the monthly instead of the daily noon time mean OH concentration, can this create an issue in the weekly cycle investigation as the monthly mean is dominated by weekdays?

Response: Thank you for this comment. We don't think that the monthly OH concentration will affect the weekly cycle of $NO_x$ emissions. First, we did not set the OH concentration fixed, it was allowed to vary within a certain range. Second, the observation of $NO_2$ and $O_3$ concentrations also reveal no significant weekdays and weekends difference, thus we could say that the OH concentration does not change much from weekdays to weekends. We have added this information in Page 16, Line 403-405 in the revised manuscript.

In the revision of the work, we discard the chemical transport model simulated OH concentration, instead we use an initial guess for the $NO_2$ chemical lifetime of 2 h for warm months (April to September) and 4 h for code months (October to March), and we let it vary between 1/4 to 4 times of the initial value during the fitting. Also, we don't think the initial guess of the $NO_2$ chemical lifetime would affect the weekly cycle of estimated $NO_x$ emissions for the same reasons we stated above.

L178: You write clear skies here and also at other parts of the manuscript, but you mean for cloud radiance fractions $< 0.5$, which means there is not always clear sky.

Response: Thank you for this comment and we have replaced the 'clear sky' with 'full-$NO_2$-coverage' in the revised manuscript.

L179: You remove overpasses with inhomogeneous wind fields and days with estimated NO$x$ emissions beyond $0.5-1.5$ times the ABACAS bottom-up emissions. First, I don't understand the reason for the ABACAS filter, and second, can you mention how many days or overpasses are filtered for each of the filters? You only mention how much it is in total. Do you maybe filter high emission days, especially in winter, which results in dampened seasonal patterns?

Response: Thank you very much for the insightful comment. The day to day, monthly and year to year variation and the uncertainty of the $NO_x$ emissions were all within the ±50% range, so the estimated data beyond this range were excluded. The extremely high and low estimation is mainly caused by the bias of the satellite $NO_2$ observation. Overall we obtained 24 days higher than 1.5 times of the a priori emission and 15 days lower than 50% of the a priori emission, the total of them made up about 10% of the total number of the estimations.

Most of the high emission days are in winter and low emission days are in summer, so we agree with the reviewer that the filter of them will dampen the seasonal variability. Thereby, in the revised work, we choose to keep these results for analysis, and we obtain a lower summer-to-winter emission ratio of 0.77 compared to the 0.87 in the initial work.

L197: Why was the area changed between Zhang et al. (2023) and this study?

Response: We narrow down our study domain from the administrative area of Wuhan in Zhang et al. (2023) to the urban area (within the Fourth Ring Road of Wuhan, ~90 km in diameter) in this study considering that, first, most (~60%) of the $NO_x$ emissions are concentrated in the urban area; second, we use regional mean wind fields and $NO_x$ chemical loss rate, the larger study domain would induce large uncertainty in the result. We have added this information in Section 2.4 in Page 5, line 147-150 in the revised manuscript.

L201: Change November 2020 to January 2020.

Response: Changed.

Figure 2 and text: y-labels say "noontime NOx emissions", which is true for TROPOMI, but I think not for EDGAR and ABACAS. Please correct this and also clarify in the text that the emission inventory emissions are not around noon time.

Response: Thank you for the comment. Yes, the bottom-up emission inventories give monthly total emissions, and when we compared them with our estimation, we converted the monthly total to hourly mean through: $\text{hourly mean emission} = \frac{monthly\ total}{(number\ of\ days\ in\ the\ month)*24}$

Then, a scale factor of 1.4 (provided by the ABACAS inventory) is applied to the hourly mean emissions to obtain the 'noontime $NO_x$ emissions'. We have made it clear in Page 8, Line 213-215 in the revised manuscript.

L240: Possible explanations for deviations with Lange et al. (2022): Larger area in Lange et al., this study limited to city center, maybe different behaviors and sources in the different urban/suburban areas? See comment L166, monthly OH concentrations dampening weekly cycle? Fixed NOx/NO2 conversion factor in this study compared to daily conversion factors in Lange et al.

Response: Thank you for the comments. We have added the discussion about the influence of the area difference on the deviations between this study and Lange et al. (2022). Please refer to Page 10, Line 257-260 in the revised manuscript.

We find from literature that the observation of $NO_2$ and $O_3$ concentrations also reveal no significant weekdays and weekends difference (Wei et al., 2022; Yang et al., 2020), thus we could say that the OH concentration and the fixed $NO_x/NO_2$ ratio do not change much from weekdays to weekends.

Section 3.2.2 Seasonal pattern: You see a much more dampened seasonal emission pattern than Lange et al. Possible explanations for deviations:

Response: Two possible factors may contribute to the large difference in the summer-to-winter emission ratio between this study and Lange et al. (2022). First is the different treatment to the $NO_x$-to-$NO_2$ ratio. We use a fixed NOx-to-NO2 ratio of 1.26, while Lange et al. (2022) calculated the ratio from day to day, and it was lower in summer than in winter, leading to a lower NOx emission estimation in summer. Second is that we use the bottom-up emission inventory to constrain our estimation, the flat seasonality of the bottom-up emissions leads to a higher summer-to-winter ratio of this study.

We have added this information in Page 11, Line 276-280 in the revised manuscript.

See comment above: You use a fixed value for the rate constant k' between NO2 and OH. How large is the seasonality due to the temperature dependency of this reaction? How large is the seasonal variation of the NOx/NO2 ratio over the year? Different areas, sampling issues due to a shorter period in Lange et al., different TROPOMI NO2 product versions have seasonal bias differences (van Geffen et al. 2022), bias correction used in this study.

Response: Thank you for the comment. According to Burkholder et al. (2020), the $NO_2$ chemical loss rate constant increases with temperature: $k'(T) = 2.8 \times 10^{-11} \times \frac{T}{300}$. In Wuhan's case, the seasonal variation of $k'$ would be ~8% (the temperature record shows that in 2023 the lowest monthly mean noon temperature is in February of 11°C, and the highest is 34°C in July); Based on the GEOS-Chem model simulation, the seasonal variation of $NO_x$/$NO_2$ ratio is ~10%.

The above two factors would lead to higher summer-to-winter emission ratio of this study compared to Lange et al. (2022). We have added this comparison in Page 11, line 276-280 in the revised manuscript.

The different study area and different study period lengths may also affect the different finding in the weekly cycle between the two studies, and we have added this discussion in Page 10, Line 257-260 in the revised manuscript.

This study uses the TROPOMI v2.4.0-2.6.0 and Lange et al. (2022) used earlier v1.x. According to Van Geffen et al. (2022), the v2.3 and later versions produce higher $NO_2$ columns over polluted area and in winter. However, this does not support the higher summer-to-winter emission ratio in this study.

We use a fixed scale factor 1.2 to correct the potential underestimation of TROPOMI $NO_2$ in this study, but the v2.4.0-2.6.0 data is still found to be lower in polluted area and higher in clean area, when compared to the ground measurements (Keppens and Lambert, 2023). Thereby our work may have overestimated the summer-to-winter emission ratio (0.77), though it is even higher in the bottom-up emission inventory. This discussion is added to Page 16, Line 408-412 in the revised manuscript.

L263/264: Any ideas for the differences in summer lifetime between Zhang et al. (2023) and this study?

Response: The possible reason for the shorter summer $NO_x$ lifetime estimated in this study than Zhang et al. (2023) can be that we use the bottom-up emission inventory to constrain our fitting, thus resulting in higher summer $NO_x$ emissions and lower chemical lifetime.

In the revision of the work, we use the result from Zhang et al. (2023) as the initial guess of the chemical loss rate of $NO_2$ during cold and warm months to replace the model simulated OH concentration.

L272: You write that the computation of NOx emissions is restricted by a priori emissions. Is this restriction dampening your estimated emissions and a possible explanation for differences with Lange et al.? See also L179 (comment above) and L388-390 in your manuscript.

Response: Yes, like the fixed $NO_x/NO_2$ ratio we use in the superposition column model, the fact that we use the bottom-up emissions to restrict our emission estimation also leads to higher summer-to-winter emission ratio estimated by our study than Lange et al. (2022). We have clarified this in the revised manuscript in Page 11, Line 276-280.

In addition, we use a fixed scale factor 1.2 to correct the potential underestimation of TROPOMI $NO_2$ in this study, but the v2.4.0-2.6.0 data is still found to be lower in polluted area and higher in clean area, when compared to the ground measurements (Keppens and Lambert, 2023). Thereby our work may have overestimated the summer-to-winter emission ratio (0.77), though it is even higher in the bottom-up emission inventory. This discussion is added to Page 16, Line 405-412 in the revised manuscript.

L298: Any ideas for large deviations between Lonsdale and Sun (2023) and this study, which kind of method is used in Lonsdale and Sun (2023)?

Response: Yes, we find that when we compared $NO_x$ emissions from two different years, we simply computed the mean from all valid days within each year. However, considering the large seasonal difference of estimated $NO_x$ emissions with the satellite data, the estimated annual mean emission will bias low when the valid summer days are more than winter days, and bias high when otherwise. Thereby, in the revision of the work, following what Lonsdale and Sun (2023) did, when we make the comparison between two years, the annual mean emissions are calculated based on the months available for both years. For example, January and February estimations are absent for 2019, June and July are not available for 2020, therefore March to May and August to December monthly values are used to determine the annual mean emissions for the comparison of the two years. As a result, the estimated year-to-year

variations in these two studies are close to each other. Please refer to Page 12, Line 311-320 in the revised manuscript.

Section 3.3 Wind field dependence: You start using the term chemical lifetime in this section, I think until now you only used the term lifetime. How do you calculate lifetimes? Usually, lifetimes estimated by these kinds of models are defined as effective lifetimes (see, e.g., Beirle et al.) as they include effects of deposition, chemical conversion, and wind advection, which you are illustrating here also for your model. I think you should clarify the differences between chemical and effective lifetime in your text.

Response: Thank you for the comment. The 'lifetime' mentioned in this study should be the 'chemical lifetime' ($\tau$, h). In the superposition model, we assume a first-order chemical loss of NO2 with the loss rate of $k$ (s$^{-1}$). the relationship between $k$ and NO$_2$ chemical lifetime $\tau_{[NO_2]}$ (h) is $k = \dfrac{1}{\tau_{[NO_2]}*3600}$. We have clarified this in the revised manuscript, please refer to Page 6, Line 164-165 in the revised manuscript.

Figure 7 and Lines following L 329: I have issues understanding this figure and the text. Explanations for what is visible in the Figure are missing in the caption and also in the text below. Is this one day, averaged for several days? How are the red, yellow, green, pink, and purple curves determined? The text in L332 says that the fitted emissions are basically in line with those from the bottom-up emissions, but in the figure differences are quite large, especially close to the city center.

Response: Thank you for the comment. As we mentioned in Section 2.4, in our fitting approach, we assume that the NO$_x$ emissions within the city is distributed as a Gaussian shape along with the wind direction: $E_i = amp * \exp[-(x_i - cen)^2/wid]$, the parameter $cen$ is initially set as the location of the city center, and we allow it to shift along the wind.

The original Figure 7 displays the estimated distribution of NO$_x$ emissions along with the wind direction. All the results are divided into four groups of the westerly, easterly, northerly and southerly wind directions. The lines in Figure 7a and b represent the mean emissions of all the days under each wind direction. We find mirrored distribution of the estimated NO$_x$ emissions between the westerly and easterly wind directions and between the southerly and northerly wind directions. The final fitted distribution is much shallower than that of the bottom-up emission inventory.

In the revision of the work, we find that this part of analysis is nothing new or different compared to the work of Lorente et al. (2019), so we chose to remove this part in the revised manuscript.

L357: Lange et al. are not using OH concentrations for their estimates and are not providing an uncertainty for OH.

Response: Thank you for the comment, since we have discarded the model simulate OH in our method, this part has been removed in the revised manuscript.

**Technical corrections:**

L47: Divide the sentence into two sentences. "….ultraviolet/visible spectrum. Various satellite instruments…"

Response: Done

L50: Misleading, change to something like "Limited by the coarse spatial resolution of the early instruments, researchers …"

Response: Done.

L55: when the target is an individual city.

Response: Corrected.

L61: Change model to method.

Response: Changed.

L63: To avoid confusion, I would change it to "The EMG model has been first applied to OMI NO2 data…"

Response: Changed.

L70: Change to: "for daily NOx emission estimates"

Response: Changed.

L72: Change to: "avoids the bias caused by using the averaged NO2 columns in the nonlinear system"

Response: Changed.

L81: Change conclusion to concluding

Response: Done.

L86: I suggest replacing "TROPOMI observes NO2 at 405-465nm of the UV-visible spectral band…" maybe with something like "TROPOMI NO2 columns are retrieved in the spectral range from 405-465nm…"

Response: Done.

L95/96: Change DLER to "the DLER".

Response: Done.

L96/97: Check the sentence starting with "The v2.4.0 version." The version is twice, and in general, it is not good to read.

Response: We rephrased this sentence to "The version 2.4.0 made a complete mission reprocessing from 1 May 2018 to 22 July 2022 and then switched to the offline mode."

L125: Change to: An initial guess…

Response: Done.

L161: Missing word: and added/combined/… with the contribution from the background

Response: we use 'combined' here.

L173: I suggest changing "dilute" with "reduce."

Response: changed.

L202: an ad hoc bias correction factor

Response: corrected.

L210/211: I would first name EDGAR, then ABACAS, following the logic of Fig. 2.

Response: Corrected. We now use EDGAR v8.1 and MEIC v1.14 for the validation.

L230: Change to "NOx emission estimates."

Response: Changed.

L237: Replace "minimum" with "reductions" and add "for Wuhan"

Response: Replaced and added.

Fig 3 caption: "The number of valid measurement days for each day of the week is listed in the plot"

Response: Corrected.

L260: Do you mean dominated instead of determined?

Response: Yes, it should be 'dominated'.

L279: Split into two sentences: "…2024a). We find a similar…"

Response: Done.

L280: Change "dramatic changes" to "strong reductions"

Response: Changed.

L283: You write "We have also found" and give a reference to Zhang et al. (2023) at the end of the sentence. Do you mean "Zhang et al. (2023) have also found"?

Response: It is a mistake, and the reference is deleted here.

Caption Figure 5: Two times NOx in the first sentence. Change dash to dashed. Split the second sentence into two sentences.

Response: Done.

[revised manuscript text omitted]

---

## Referee Report (RR1)

Review of the manuscript: "Estimating the variability of NOx emissions from Wuhan with TROPOMI NO2 data during 2018 to 2023" by Zhang et al.

I would recommend accepting the manuscript for publication after minor revisions as follows (the line numbers refer to the revised manuscript):

L28 rephase as: "though it is even higher in the bottom-up inventories"

L33 ", the estimation for Wuhan is ~4% for the emissions and ~8% for the chemical lifetime." What is the estimation here? Do you mean the underestimation compared to something? Not sure what these numbers represents... Please clarify.

L146 "114.4°E, 30.6°N," put this in parenthesis maybe?

L147- "Compared to Zhang et al. (2023), our study domain is limited to the urban area (within the Fourth Ring Road) of Wuhan. For one reason, most (~ 60%) of the NOx emissions are concentrated in the urban area (Zhang et al., 2023); for another, we use regional mean wind fields and NOx chemical loss rate, the larger study domain would induce large uncertainty in the result."
This sentence is a bit strangely formulated. Maybe try to rephrase it for example as follows: "Compared to Zhang et al. (2023), our study domain is limited to the urban area (within the Fourth Ring Road) of Wuhan, as most (~ 60%) of the NOx emissions are concentrated in this area (Zhang et al., 2023). In addition, since we use regional mean wind fields and NOx chemical loss rate, the larger study domain would induce larger uncertainty in the result." Or something similar.

L151 "One demission of the grid map is along with and the other perpendicular to the wind direction." Maybe rephrase as "One dimension of the grid map is along the wind direction, and the other is perpendicular to it."

L153 "rotated grid map" should you say here as well resampled instead of rotated? Same in the title of Fig. 1b.

L229-231 "Overall we calculate a mean NOx chemical lifetime of 2.82 h, close to the 2.46 h estimated by Zhang et al. (2023), and is around 5% lower than Lange et al. (2022) reported 2.94±0.3 h for the NOx effective lifetime." Rephrase maybe as follows: "Overall we obtain a mean NOx chemical lifetime of 2.82 h, which is close to the 2.46 h estimated by Zhang et al. (2023), and around 5% lower than the value (2.94±0.3 h) reported by Lange et al. (2022) for the NOx effective lifetime."

L231 "The fitting result for cold months NOx chemical lifetime is 4.25 h, and for most of the days, the estimated NOx chemical lifetime is between 1.5 h and 6 h." replace with "For the cold months the estimated NOx chemical lifetime is 4.25 h, and for most of the days, the estimated NOx chemical lifetime is between 1.5 h and 6 h." Also what do you mean with "most of the days"? How many?

L313 here and elsewhere, the word "valley" could be replaced with "sudden decrease/reduction/drop" or something similar.

L337 we compute -> we obtain

L398 we discard, We should be with first capital letter

L402 "Our analysis of the temporal variability of the estimated NOx chemical lifetime and emission is also of uncertainty, though this part of uncertainty is difficult to quantify." This sentence does not read right, please check and rephrase.

L404 ... for this is also found... you mean: "since this.."?

L405 "Second, we have overestimated the summer-to-winter emission ratio..." Overestimated compared to what?

L422 align -> aligned

L446 "dominated with fast winds." -> dominated by fast winds

Overall, I would recommend a review by an English mother-tongue, because I might not be able to catch imprecisions in English grammar.

---

## Author Response (AR2)

Dear Editor,
The authors thank you and the reviewers for the efforts you and the reviewers put into improving this manuscript. The point-by-point response to your comments is listed below and the revisions/additions/edits are shown in the tracked-change file.

I'm please to accept your manuscript "Estimating the variability of NOx emissions from Wuhan with TROPOMI NO2 data during 2018 to 2023" after minor revisions. Two of the reviewers have provided lists with minor comments - please carefully address them in your revised manuscript. In addition, please check my comments below.

* Abstract: I suggest to reformulate the sentence "We estimate a summer-to-winter emission ratio of 0.77, which is overestimated to some extent, though it is even higher provided by the bottom-up inventories" to "We estimate a summer-to-winter emission ratio of 0.77, which may be overestimated to some extent, but is still lower than suggested by bottom-up inventories"

Response: Done.

* Figure 1: Wrong units in emissions in right figure

Response: Corrected.

* Figure 4. Please add label to y-axis even if the quantity shown is the normalised emission

Response: Added.

Review #1:
Review of the manuscript: "Estimating the variability of NOx emissions from Wuhan with TROPOMI NO2 data during 2018 to 2023" by Zhang et al.
I would recommend accepting the manuscript for publication after minor revisions as follows (the line numbers refer to the revised manuscript):

L28 rephase as: "though it is even higher in the bottom-up inventories"

Response: This sentence has been rephrased as 'We estimate a summer-to-winter emission ratio of 0.77, which may be overestimated to some extent, but is still lower than suggested by the bottom-up inventories.'

L33 ", the estimation for Wuhan is ~4% for the emissions and ~8% for the chemical lifetime." What is the estimation here? Do you mean the underestimation compared to something? Not sure what these numbers represents… Please clarify.

Response: It should be 'underestimation'. This sentence is rephrased as '…in Wuhan's case, the underestimation is ~4% for the emissions and ~8% for the chemical lifetime'.

L146 "114.4°E, 30.6°N," put this in parenthesis maybe?

Response: added.

L147- "Compared to Zhang et al. (2023), our study domain is limited to the urban area (within the Fourth Ring Road) of Wuhan. For one reason, most (~ 60%) of the NOx emissions are concentrated in the urban area (Zhang et al., 2023); for another, we use regional mean wind fields and NOx chemical loss rate, the larger study domain would induce large uncertainty in the result." This sentence is a bit strangely formulated. Maybe try to rephrase it for example as follows: "Compared to Zhang et al. (2023), our study domain is limited to the urban area (within the Fourth Ring Road) of Wuhan, as most (~ 60%) of the NOx emissions are concentrated in this area (Zhang et al., 2023). In addition, since we use regional mean wind fields and NOx chemical loss rate, the larger study domain would induce larger uncertainty in the result." Or something similar.

Response: This sentence is rephrased as suggested.

L151 "One demission of the grid map is along with and the other perpendicular to the wind direction." Maybe rephrase as "One dimension of the grid map is along the wind direction, and the other is perpendicular to it."

Response: Rephrased.

L153 "rotated grid map" should you say here as well resampled instead of rotated? Same in the title of Fig. 1b.

Response: The grid map is rotated, and the TROPOMI observation is resampled into the rotated grid map. We have corrected the title of Fig. 1b.

L229-231 "Overall we calculate a mean NOx chemical lifetime of 2.82 h, close to the 2.46 h estimated by Zhang et al. (2023), and is around 5% lower than Lange et al. (2022) reported 2.94±0.3 h for the NOx effective lifetime." Rephrase maybe as follows: "Overall we obtain a mean NOx chemical lifetime of 2.82 h, which is close to the 2.46 h estimated by Zhang et al. (2023), and around 5% lower than the value (2.94±0.3 h) reported by Lange et al. (2022) for the NOx effective lifetime."

Response: Rephrased.

L231 "The fitting result for cold months NOx chemical lifetime is 4.25 h, and for most of the days, the estimated NOx chemical lifetime is between 1.5 h and 6 h." replace with "For the cold months the estimated NOx chemical lifetime is 4.25 h, and for most of the days, the estimated NOx chemical lifetime is between 1.5 h and 6 h."

Also what do you mean with "most of the days"? How many?

Response: The sentence is rephrased as: 'For the cold months, the estimated NOx chemical lifetime is 4.25 h, and for ~70% of the days, the estimated NOx chemical lifetime is between 1.5 h and 6h. For the warm months, ~65% of the chemical lifetime estimation is within the 0.8 – 2.5 h range, and the mean value is 1.62 h.'

L313 here and elsewhere, the word "valley" could be replaced with "sudden decrease/reduction/drop" or something similar.

Response: Corrected.

L337 we compute -> we obtain

Response: Corrected.

L398 we discard, We should be with first capital letter

Response: Corrected.

L402 "Our analysis of the temporal variability of the estimated NOx chemical lifetime and emission is also of uncertainty, though this part of uncertainty is difficult to quantify." This sentence does not read right, please check and rephrase.

Response: This sentence is changed to 'Uncertainties also exist in the analysis of the temporal variability of the estimated $NO_x$ chemical lifetimes and emissions.'

L404 … for this is also found… you mean: "since this.."?

Response: Corrected.

L405 "Second, we have overestimated the summer-to-winter emission ratio…" Overestimated compared to what?

Response: During the fitting procedure, we use a fixed $NO_2$-to-$NO_x$ ratio and the bottom-up emissions to constrain the NOx emissions, we may have obtained a dampened seasonality of $NO_x$ emissions, thus a higher summer-to-winter emission ratio. Therefore, we say that 'we have overestimated the summer-to-winter emission ratio'.

L422 align -> aligned

Response: Corrected.

L446 "dominated with fast winds." -> dominated by fast winds

Response: Corrected.

Overall, I would recommend a review by an English mother-tongue, because I might not be able to catch imprecisions in English grammar.

Response: Thank you for the recommendation. A mother-tongue of English was invited to review the grammar.

Review #2:

Dear Zhang et al.,

Thank you for the updated version of the manuscript and the detailed answers to the review comments. Most of the suggestions from the reviewers are considered, which improves the quality of the manuscript and understanding of the study.

However, I still have some concerns about certain parts of the methodology and the resulting seasonality of the estimated emissions. The fixed NOx/NO2 ratio, the fixed bias correction factor, and constraining the fitting based on bottom-up emissions with a flat seasonality all lead to a dampened seasonality in the estimated emissions. These points are more discussed in the new version of the manuscript but must be kept in mind when analyzing variability.

The authors followed the reviewer's suggestions to remove the filter, which filtered days with estimated emissions beyond 0.5-1.5 times the ABACAS bottom-up emissions; another point in the method that dampened the seasonality. This results in a lower summer-to-winter emission ratio of 0.77 compared to 0.87 in the original manuscript.

The authors followed the suggestion to compare their estimated emissions to the monthly emissions of the EDGAR emission inventory. This provides new/more insights into the seasonal deviations between bottom-up and top-down emission estimates. With, in general, lower emissions from the top-down approach but much closer agreements in winter and larger discrepancies during the summer months.

In the revision, the term lifetime was changed to chemical lifetime. Can your estimated lifetimes represent the true chemical lifetime? Since it is usually influenced by downwind changes (influencing additional sources, plume meandering, wind), it is often called mixed or effective lifetime (e.g., Valin et al. 2015, Beirle et al. 2011, supplement).

Response: Thank you for the comments. The $NO_x$ lifetime estimated through the superposition column model is derived from the $NO_x$ chemical loss rate ($\tau = \frac{1}{k}$), and this is why it is referred to as the 'chemical lifetime'. The chemical lifetime is a part of the 'effective lifetime', which is composed of the chemical loss, the plume meandering, and other loss pathways.

Minor comments:

Line 99: Change "Compared to the previous versions v1.x, the version 2.3.1 includes a different treatment of the surface albedo to avoid negative and > 1 cloud fractions, and updates the FRESCO-wide cloud retrieval that leads to a lowering cloud pressure. " to "Compared to the previous versions v1.x, version 2.3.1 includes a different treatment of the surface albedo to avoid negative and > 1 cloud fractions, as well as an updated

FRESCO-wide cloud retrieval resulting in lower cloud pressures."

Response: Changed.

Line 167: You don't say what your initial guess of 2h and 4h lifetime for summer and winter is based on. Later, in line 226, you say it is based on Zhang et al. 2023 but don't mention the details anymore. Please add information in Line 167 and maybe again in line 226.

Response: Added.

Line 178: I think the second part of the sentence "for too large a varying range is applied for it." doesn't make sense like this. For too large what? Instead of "a varying range is applied for it" maybe "a scale factor (fac) is applied to it"

Response: This sentence is rephrased as 'The emission term is used in the cost function to reduce the dependence of fitted $NO_x$ chemical lifetimes and emissions on the $\tau_{[NO_2]}$.'

Line 205: Change ", and they …" to ". However, they …"

Response: Done.

Line 218: I think comparing the monthly emission inventory data and the different deviations in summer and winter is interesting. Can you comment on possible reasons?

Response: When we use a top-down method to estimate NOx emissions, we assume that the atmosphere is in a state of equilibrium: Emission – Loss = Atmospheric Content. In cold months, because of the slow loss rate, it is easy to reach equilibrium. However, in warm months, especially in summer, the fast loss of $NO_x$ may make it difficult to achieve the equivalent state. Thus it might lead to underestimation of NOx emissions of the top-down method in warm months.

Line 226: is determined instead of are determined

Response: Corrected.

Line 321: Delete significant. "A decrease (13.6%) is seen in 2023 compared to 2022."

Response: Deleted.

Line 349: Replace "our" with "the"

Response: Done.

Line 359: "…the superposition column underestimate…", the word model is missing here

Response: Added.

Line 405-410: Please add the point you mentioned in Sect. 3.2.2 here again; you are constraining your fit with emission inventory data, which shows a flat seasonality, probably leading to a dampened seasonality in your estimates.

Response: Added.

Line 429: Is the 2018-2019 comparison for the MEIC emission inventory? I think this information is missing here.

Response: The sentence is rephased as 'TROPOMI estimation is lower than MEIC by less than 30% for 2018−2020. When compared with EDGAR v8.1, TROPOMI estimation is lower by ~20% for 2018 and 2019, but is 30~40% lower in 2020−2022.'